# Evaluation and comparison of methods for neuronal parameter optimization using the Neuroptimus software framework

**Máté Mohácsi[1,2], Márk Patrik Török[1,2], Sára Sáray[1,2], Luca Tar[1,2], Gábor Farkas[1,2], Szabolcs Káli** [1,2] *

**1** HUN-REN Institute of Experimental Medicine, Budapest, Hungary, **2** Faculty of Information Technology and Bionics, Pázmány Péter Catholic University, Budapest, Hungary

* kali.szabolcs@koki.hun-ren.hu

**Data Availability Statement:** The source code of our software tool Neuroptimus, as well as the code required to run all of our use cases is available at

## Abstract

Finding optimal parameters for detailed neuronal models is a ubiquitous challenge in neuro-scientific research. In recent years, manual model tuning has been gradually replaced by automated parameter search using a variety of different tools and methods. However, using most of these software tools and choosing the most appropriate algorithm for a given optimization task require substantial technical expertise, which prevents the majority of researchers from using these methods effectively. To address these issues, we developed a generic platform (called Neuroptimus) that allows users to set up neural parameter optimization tasks via a graphical interface, and to solve these tasks using a wide selection of state-of-the-art parameter search methods implemented by five different Python packages. Neuroptimus also offers several features to support more advanced usage, including the ability to run most algorithms in parallel, which allows it to take advantage of high-performance computing architectures. We used the common interface provided by Neuroptimus to conduct a detailed comparison of more than twenty different algorithms (and implementations) on six distinct benchmarks that represent typical scenarios in neuronal parameter search. We quantified the performance of the algorithms in terms of the best solutions found and in terms of convergence speed. We identified several algorithms, including covariance matrix adaptation evolution strategy and particle swarm optimization, that consistently, without any fine-tuning, found good solutions in all of our use cases. By contrast, some other algorithms including all local search methods provided good solutions only for the simplest use cases, and failed completely on more complex problems. We also demonstrate the versatility of Neuroptimus by applying it to an additional use case that involves tuning the parameters of a subcellular model of biochemical pathways. Finally, we created an online database that allows uploading, querying and analyzing the results of optimization runs performed by Neuroptimus, which enables all researchers to update and extend the current benchmarking study. The tools and analysis we provide should aid members of the neuroscience community to apply parameter search methods more effectively in their research.

https://github.com/KaliLab/neuroptimus. We also created a website (https://neuroptimus.koki.hu/) where we make available the detailed results of all the optimization runs that are included in the results of this paper. We also uploaded all models, scripts, and simulation results to a public data repository (https://doi.org/10.5281/zenodo.13918521).

**Funding:** This project received funding from the European Union's Horizon 2020 Framework Programme for Research and Innovation under Specific Grant Agreements No. 720270 and No. 785907 (Human Brain Project SGA1 and SGA2; SK), and from the European Union project RRF-2.3.1-21-2022-00004 within the framework of the Artificial Intelligence National Laboratory (SK). The funders had no role in study design, data collection and analysis, decision to publish, or preparation of the manuscript.

**Competing interests:** The authors declare that no competing interests exist.

## Author summary

Model fitting is a widely used method in scientific research. It involves tuning the free parameters of a model until its output best matches the corresponding experimental data. Finding the optimal parameter combination can be a difficult task for more complex models with many unknown parameters, and a large variety of different approaches have been proposed to solve this problem. However, setting up a parameter search task and employing an efficient algorithm for its solution requires considerable technical expertise. We have developed a software framework that helps users solve this task, focusing on the domain of detailed models of single neurons. Our open-source software, called Neuroptimus, has a graphical interface that guides users through the steps of setting up a parameter optimization task, and allows them to select from more than twenty different algorithms to solve the problem. We have also compared the performance of these algorithms on a set of six parameter search tasks that are typical in neuroscience, and identified several algorithms that delivered consistently good performance. Finally, we designed and implemented a website that allows users to view and analyze our results and to add their own results to the database.

## Introduction

The construction and simulation of data-driven models has become a standard tool in neuroscience [1–3]. Such models can be employed, among other things, to consolidate the knowledge obtained from various experimental approaches into a common framework, to test the consistency of the data, and to make novel predictions by examining the response of the model to arbitrary inputs and by applying clean manipulations. Models at a given level of description (e.g., individual neurons) can also be combined to form models of entities at higher levels (such as networks) and thus aid the mechanistic understanding of emergent phenomena.

Nevertheless, these data-driven models often contain parameters that are not directly constrained (or are only weakly constrained) by the available experimental data. Traditionally, such unknown parameters were often tuned manually to adjust the behavior of the model towards some desired target. However, this approach is typically inefficient, not quantitative, and may be heavily biased to reproduce a few selected experimental results at the expense of other relevant data. Consequently, in recent years, automated parameter search has emerged as the preferred method for the estimation of unknown parameters of neural models [4–20]. This approach requires the definition of an error function (or cost function) that measures the quality of the model with a given set of parameters, often in terms of how well it approximates data obtained using a particular experimental protocol. The goal of parameter optimization is then to find the set of parameters that minimizes the selected cost function. The difficulty of this task can vary widely depending on the nature and complexity of the model, the definition of the error function (or multiple error functions representing different goals, or objectives), and the number of unknown parameters. Simple optimization problems can be solved effectively by traditional gradient-based, local methods or by random search, but these approaches tend to fail when there are many unknown parameters and the cost function has multiple local minima [15,21]. In fact, no algorithm is guaranteed to find the globally optimal parameter combination in a short time for all problems [22], and various clever search methods (called metaheuristics) have been proposed that often find good solutions in an acceptable amount of time by taking advantage of various types of regularities in the cost function [23].

Previous studies in neuroscience have used a variety of different software tools and algorithms to perform parameter optimization. The general-purpose neural simulators NEURON [24] and GENESIS [25] both include implementations of a few selected methods that are adequate for certain parameter search tasks. In addition, several tools have been developed specifically for neural parameter optimization, including Neurofitter [26], BluePyOpt [27], pypet [28], and NeuroTune [29], and some more general computational neuroscience tools such as NetPyNE [30] also have some support for parameter optimization. However, most of these tools rely on a very limited set of parameter search methods, which typically does not include many optimization algorithms that represent the state of the art in global optimization and are popular in other fields of science and engineering. These new methods were not included in any previous surveys of neural optimization. Systematic comparisons of the existing neural optimization software tools have also been quite limited [15]. Therefore, it is currently unknown which parameter search methods can be expected to perform well in the parameter optimization tasks that are typical in neuroscience.

Furthermore, most of the existing tools for neural optimization lack any intuitive user interface, and require substantial programming experience. One exception is our earlier optimization software called Optimizer [15], which included a graphical user interface (GUI) that was designed to guide users through the process of setting up, running, and evaluating the results of a neuronal parameter optimization task. Optimizer also provided four different optimization algorithms in two different Python packages, and was designed in a modular way to facilitate the integration of new components including additional optimization algorithms. However, Optimizer has become outdated in the ten years since its original publication [15], partly because it was written in version 2 of Python that is not supported any more, and because it had no support for parallelization, which limited its usefulness in more complex optimization problems. Overall, although a wide variety of research projects require finding the optimal parameters of neural models, no currently available tool allows researchers with limited computational expertise to set up, run, and evaluate neural parameter optimization tasks using the best known algorithms.

The goal of the current study was twofold. First, we aimed to provide a general software framework that allows the straightforward application of a large variety of state-of-the-art parameter optimization methods to typical problems in data-driven neural modeling. This was accomplished by significantly updating and extending our software tool (which is now called Neuroptimus). Second, we aimed to perform a systematic comparison of parameter search methods (including both previously used and novel algorithms) in the context of modeling single neurons, which is probably the most common subtype of parameter optimization tasks in neuroscience. To this end, we designed and implemented a test suite of neuronal parameter optimization problems, and used Neuroptimus to systematically test the performance of a large set of optimization algorithms on each of these benchmarks. The results of the different algorithms on the test suite were systematically analyzed and compared. Finally, we designed and deployed a web-accessible database that contains all the results of this study and also allows users to upload, retrieve, and analyze the results of parameter optimization. By providing an accessible but also versatile software tool, clear recommendations regarding the selection of the best available optimization methods, and an extensible online database of optimization results, we hope to enable a larger portion of the neuroscientific community to use systematic parameter search effectively in their research.

## Results

The systematic evaluation of parameter optimization methods in the context of neuronal modeling required the development of several interrelated methods and tools, which are

described in detail in the Methods section and whose main features are also summarized below. The first necessary ingredient was a software tool that allows users to set up, execute, and evaluate the results of a wide variety of neural parameter optimization problems in a single standardized framework. The second required component was a diverse set of benchmark problems that differ in the type of the model, the number of unknown parameters, and the complexity of the error function, and that collectively cover many types of parameter fitting problems that are often encountered in neuronal modeling. The third necessary component was a set of methods that allows consistent evaluation and comparison of optimization results across the different benchmarks and algorithms. Finally, the last ingredient was a web-accessible database of optimization results that allows us to share all of our results publicly and also enables us as well as other researchers to extend the study with additional optimization runs and even new benchmarks.

## The Neural Optimization User Interface (Neuroptimus)

We began our study by updating, improving and extending our previously developed optimization software (Optimizer), which was already shown to be a useful tool for neuronal optimization [15]. The new version (named Neuroptimus) inherited many useful features from its predecessor, and added several important new capabilities. Both Optimizer and Neuroptimus support the definition and solution of neural optimization problems through a graphical user interface (GUI) that guides the users throughout the process. The main steps (represented by different tabs in the GUI) involve selecting the target data, selecting the model and the parameters to be optimized, setting up the simulations (including stimulation and recording parameters), defining the cost function, selecting the optimization algorithm, running the parameter search, and reviewing the results. A detailed guide to the GUI is available in the online documentation of Neuroptimus (https://neuroptimus.readthedocs.io/en/latest/). All the functionality is also accessible through a command line interface that uses configuration files to set up the optimization, which enables batch processing (e.g., multiple runs with different settings or random seeds). Simulations of the model can be performed either by the NEURON simulator [24], which is handled internally, or by arbitrary external code (which may include running other simulators) handled as a "black box". The modular, object-oriented structure of the program makes it possible to extend its capabilities by adding new error functions and optimization algorithms.

Neuroptimus includes several new and enhanced features compared to Optimizer. In addition to specific time series (such as voltage traces), it is now also possible to use as target data the statistics of features extracted (e.g., using the feature extraction module eFEL, [31]) from a set of experimental recordings. In this case, Neuroptimus uses eFEL to extract the same features from each simulated model, computes feature errors as the difference between the feature value of the model and the mean value of the experimental feature, normalized by the experimental standard deviation, and uses the sum of these feature errors as the cost function during parameter optimization. Weights can also be provided individually for each error component.

While Optimizer provided four different search algorithms (two local and two global algorithms implemented by the Inspyred and SciPy packages), Neuroptimus currently supports more than twenty different optimization algorithms from five external Python packages (see Methods for a complete list), plus an internally implemented random sampling algorithm, which can be considered as a simple baseline method. To aid novice users in the selection of appropriate optimization algorithms, the Neuroptimus GUI also provides a list of recommended algorithms and default settings.

Neuroptimus also contains many enhancements "under the hood". The new version was entirely developed in Python 3 to support recent open-source Python modules, such as search

algorithms, graphical and parallelization interfaces. The graphical user interface was completely re-implemented using the PyQt5 package, which provides a Python binding to the popular cross-platform GUI toolkit Qt. In addition to the parameter search methods offered by SciPy and Inspyred, Neuroptimus now also provides an interface to the algorithms implemented by the widely used optimization packages Pygmo and BluePyOpt, as well as an additional parallelized Python implementation of the Covariance Matrix Adaptation Evolution Strategy (CMAES) algorithm. For many of these search algorithms, parallel evaluation of models is also supported and easily configurable, which can lead to a manifold reduction in execution time, especially on highly parallel architectures such as compute clusters and supercomputers.

## Neural optimization benchmarks

We defined and implemented a test suite of different neuronal optimization problems to demonstrate the utility of our Neuroptimus software and to quantitatively evaluate and compare the effectiveness of different parameter optimization algorithms. Our aim was to identify which parameter search methods (and which implementations) are able to find good solutions to each of our benchmark problems, and which methods (if any) can provide consistently good performance across all of these tasks. Our benchmarking use cases differ in the complexity of the models, the simulation protocol, the source and nature of the target data, the features and error functions used to evaluate the model, and the number of unknown parameters. A subset of our use cases is analogous to those that were described by Friedrich et al. [15], although some of these have been updated to improve their robustness. Each of the six benchmark problems is described briefly below, and in more detail in the Methods section.

Four of the use cases involve finding the biophysical parameters of compartmental models of neurons based on somatic voltage responses; however, these models differ greatly in terms of the level of morphological and biophysical detail, and also in the number of unknown parameters (between 3 and 12). One simple use case involves the classic single-compartment Hodgkin-Huxley model with two voltage-gated conductances and a leak conductance; one uses a morphologically detailed but passive model neuron; another benchmark optimizes the somatic conductances of several voltage-gated ion channels in a simplified (6-compartment) model, while our most complex use case involves fitting spatially varying conductance densities for a large set of ion channels in a fully detailed compartmental model of a hippocampal pyramidal cell. A different type of benchmark involves optimizing the parameters of a phenomenological point neuron (an adaptive exponential integrate-and-fire model), and the final one simulates a voltage-clamp experiment to estimate synaptic parameters.

Some of our benchmark problems (the Hodgkin-Huxley and the Voltage Clamp use cases) use surrogate data as the target. In this case, target data are generated by the same neuronal model with known parameters; some of these parameters are then considered to be unknown, and the task is to reconstruct the correct values [32]. Therefore, in these test cases, a perfect solution with zero error is known to exist, and the corresponding parameters can be compared to those found by the search algorithms. However, for most of our benchmark problems, the target data were recorded in electrophysiological experiments, or (in one case) generated by a more complex model than the one we were fitting. In these instances, the best-fitting parameters and the minimal possible error score are unknown.

In most of our use cases we compared the output of the model to the target data by extracting several different electrophysiological features from the raw voltage traces. The difference of each model feature from the corresponding (mean) experimental feature can be considered as a separate error component (or objective). This allowed the direct application of multi-

objective optimization methods. When using single-objective algorithms, feature errors were combined into a single cost function using an average with pre-defined (in most cases, uniform) weights. The final best solution for multi-objective algorithms was also chosen using the same weighted average of the objectives. Two of our use cases had simpler voltage or current traces as their target. In these cases, the mean squared difference between the model trace and the experimental trace was used as the only error function. This precluded the use of multi-objective optimization methods, so only single-objective algorithms were included in the comparison in these cases.

We used several criteria to select optimization algorithms for inclusion in our benchmark study. First, we implemented a simple random search algorithm based on independently repeated uniform sampling of the entire available search space defined by the parameter boundaries. This algorithm can be considered as a natural baseline against which we can measure the performance of more sophisticated methods. Second, we included some popular local optimization algorithms (Nelder-Mead and L-BFGS-B) that are expected to be efficient when the error function has a single optimum, but not for more complex problems with multiple local optima. The rest of the search algorithms that we included are so-called global optimization methods or meta-heuristics, which aim to take advantage of certain types of regularities in the error function to find the global optimum (or another similarly good solution) more efficiently than a random search or local optimization methods do. A very large selection of such meta-heuristic algorithms has been developed, and many of these are included in one (or several) of the Python packages that are accessible in Neuroptimus. Due to time and resource constraints, not all of these algorithms were included in the current study, but we aimed to include many of the algorithms that were previously used in neuronal optimization and those that have proved particularly successful in other settings. More specifically, we included several different types of evolutionary algorithms, several implementations of the particle swarm algorithm, and also some other types of bioinspired algorithms and methods based on statistical physics.

To ensure a fair comparison of different search methods, we allowed a maximum of 10,000 model evaluations in a single run of every optimization algorithm. For all the algorithms that define populations of models that are evaluated as a batch in every iteration (this includes both evolutionary and swarm algorithms), we set the population size to 100, and ran the algorithms for 100 iterations (generations). We recorded the lowest error value achieved during each run, and also looked at how the best error score evolved during the course of the optimization. This allowed us to quantify the speed of convergence by calculating the area under the curve showing the cumulative minimum error as a function of completed model evaluations. Neuroptimus also saves all the error components for each model evaluated during the optimization, so that it is possible to trace the evolution of individual error components (see S1 Fig for an example). We performed 10 repeated runs of each algorithm on every benchmark problem to allow proper statistical evaluation of the results.

A more detailed description of the algorithms, use cases, and evaluation methods in our benchmarking study can be found in the Methods section.

## The performance of different optimization algorithms on individual benchmarks

**Use Case 1: Hodgkin-Huxley model.** Our first benchmark problem involved finding the correct densities of two voltage-gated conductances and the leak conductance (3 parameters overall) in the classic single-compartment Hodgkin-Huxley model [33] based on the voltage response to a single current step stimulus (Fig 1). We compared the response of each candidate

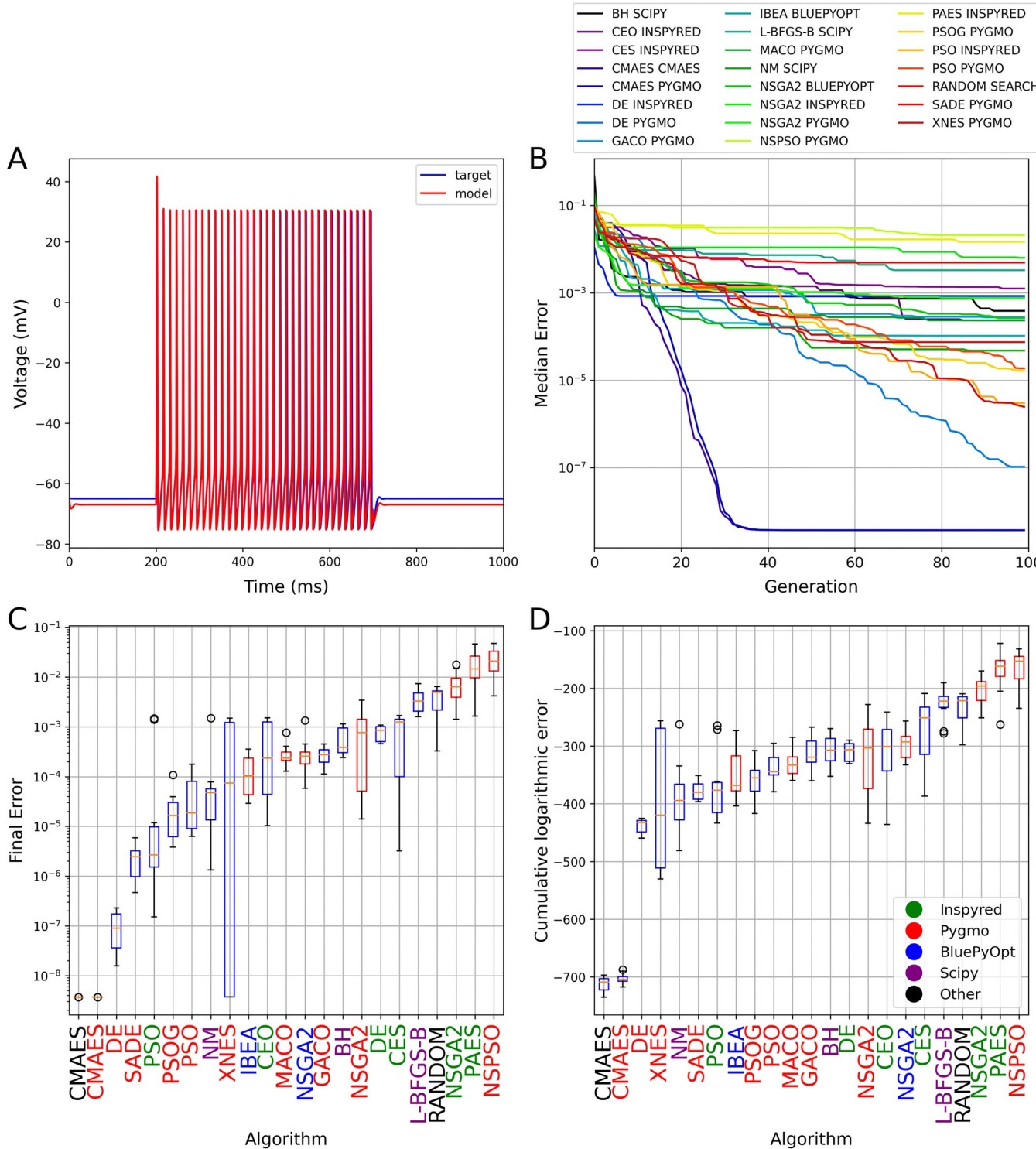

**Fig 1. The results of fitting conductance densities in the Hodgkin-Huxley model.** (A) Example of a comparison plot showing the voltage trace generated by the model with its original parameters (blue) and the trace given by the model using the best parameter set found by the Random Search algorithm (red). (B) Plot showing the evolution of the cumulative minimum error during the optimization. The curves show the median of 10 independent runs for each relevant algorithm. Each generation corresponds to 100 model evaluations. The colors corresponding to the different algorithms (and packages) are shown in the legend. (C) Box plot representing the distribution of the final error scores over 10 independent runs of each algorithm. (D) Box plot representing the convergence speed of the algorithms tested, measured as the area under the logarithmic cumulative minimum error curve (as shown in panel B). In (C) and (D), horizontal red lines indicate the median, the boxes represent the interquartile range, whiskers show the full range (excluding outliers), and circles represent

outliers. Boxes representing single-objective algorithms are colored blue and those of multi-objective ones are red. Results are sorted by the median score, from the best to the worst. The names of the packages on the horizontal axis are colored to indicate the implementing package according to the legend in (D).

model to that of the original model by evaluating four features (spike count, spike amplitude, spike width, and mean squared error of the voltage excluding spikes, evaluated using built-in error functions of Neuroptimus), which also enabled the application of multi-objective optimization methods. We expected this to be a relatively simple optimization problem based on the small number of parameters to fit, although it is also clearly non-trivial due to the nonlinear nature of the neuronal dynamics and, particularly, the complicated dependence of the extracted physiological features on the conductance parameters.

Many of the search algorithms tested found relatively good solutions most of the time (Fig 1C; see also S2 Fig for a direct comparison between the best results provided by CMAES and random search at the level of individual voltage traces), but most of them failed to converge completely in 10,000 model evaluations. The exception was the CMAES algorithm, whose implementations both consistently converged to the optimal solution after approximately 3,500 evaluations (the lowest possible error score was not exactly zero due to rounding errors). Interestingly, multi-objective algorithms generally performed worse on this use case than single-objective ones, with Inspyred's NSGA2, PAES and Pygmo's NSPSO algorithms giving worse results than Random Search. Different implementations of the same algorithms (two versions for CMAES, three for PSO, and three for NSGA2) usually showed similar convergence behavior, except for the implementation of NSGA2 by the Inspyred package that performed significantly worse than the Pygmo and BluePyOpt versions of the same method. Overall, even this simple benchmark revealed surprisingly large differences in the performance of the different search methods that we included in our comparison.

**Use Case 2: Voltage Clamp.** The second benchmark problem involved finding four parameters of a simulated synaptic connection to a single-compartment model neuron using voltage-clamp recordings (Fig 2). This use case also used surrogate data as the target, but in this case the recorded variable was the current injected by the electrode during a simulated voltage-clamp experiment. The parameters to be reconstructed were the maximal value (weight), delay, and rise and decay times of the synaptic conductance change following each repeated activation of the synapse. Due to the stereotyped nature of the data, mean squared difference was used as the only error function, and thus only single-objective algorithms were tested.

Although this is still a relatively simple and low-dimensional problem, and the intrinsic dynamics is much less complex than that of the Hodgkin-Huxley model in current clamp mode in the first use case above, we observed highly divergent performance for the set of algorithms that we tested (Fig 2C). Many of the algorithms found solutions with acceptable fits to the data (see S3 Fig for a direct comparison between the best results provided by CMAES and random search), but there were large differences in the speed of convergence to the lowest possible error score and the correct parameter values (Fig 2B and 2D). Both implementations of CMAES reached the best possible score (again defined by round-off error) in fewer than 40 generations (4000 model evaluations). The Inspyred implementation of PSO also approached this limit by the end of the optimization (10,000 model evaluations), but it converged substantially slower than CMAES. The Pygmo implementations of PSO, two versions of the DE algorithm, and the CES algorithm of Inspyred also achieved good results, but converged even more slowly. At the other end of the spectrum, local search algorithms were typically not effective at solving this problem (although they found very good solutions in some cases), and the XNES algorithm from the Pygmo package actually performed worse than the baseline random search method.

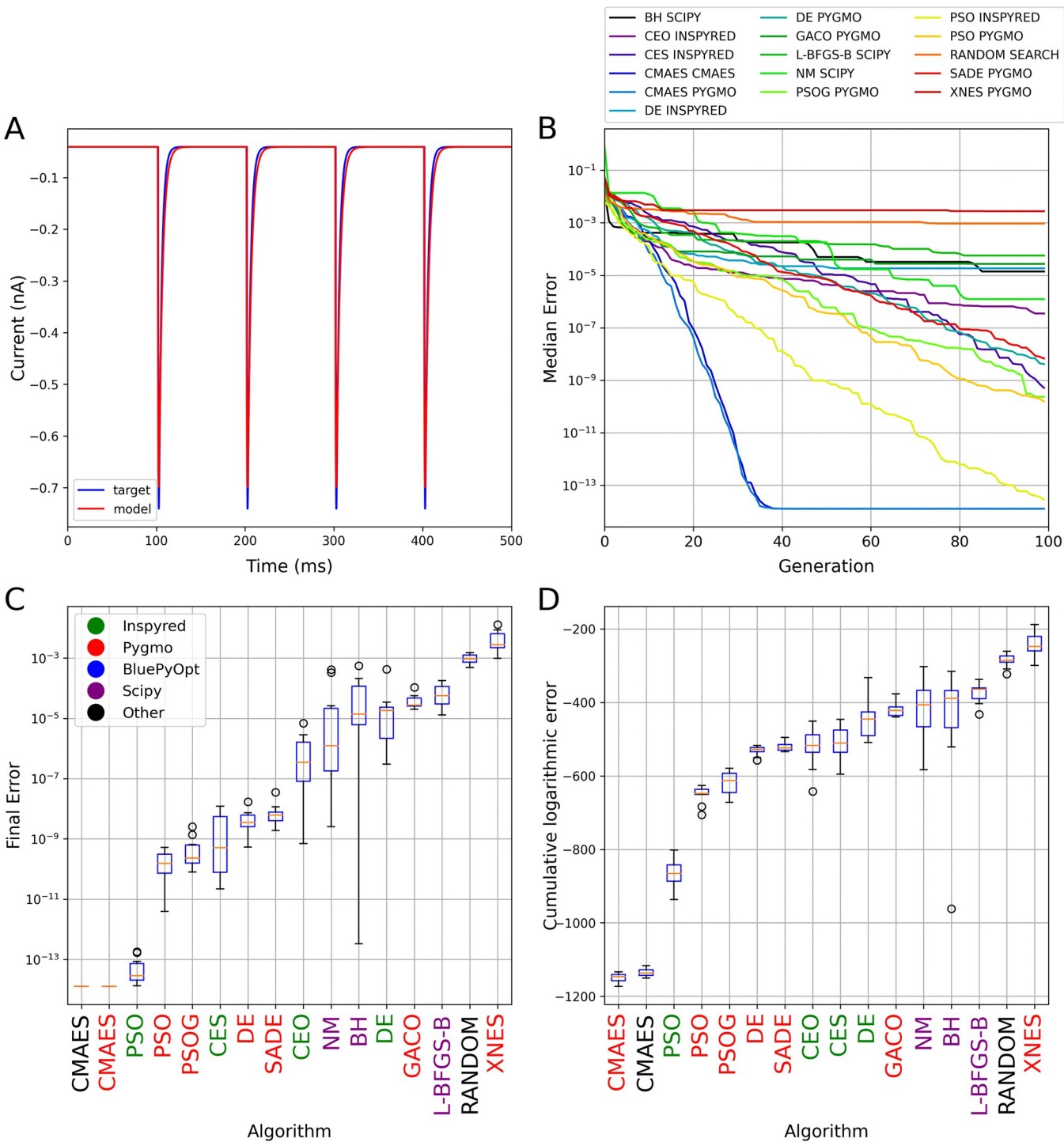

**Fig 2. The results of fitting the parameters of a synaptic connection based on simulated voltage-clamp recordings.** The plots in all four panels are analogous to those in Fig 1. Panel A shows the results of a best-fitting model found by the Random Search algorithm. Note that the error function had only a single component in this use case, and therefore only single-objective optimization algorithms were compared.

**Use Case 3: Passive, anatomically detailed neuron.** This use case represents an important practical problem that has been investigated in several previous studies [15,34–36]. It involves the estimation of three basic biophysical parameters that determine the propagation

and integration of voltage signals within neurons in the subthreshold voltage range: the (specific) membrane capacitance, membrane resistance, and axial resistance. The task is to estimate these three parameters based on the voltage response of a neuron to a current stimulus (which, in our case, consisted of a larger short and a smaller long current step) recorded from a hippocampal pyramidal cell in vitro (Fig 3). The response of the model is linear in terms of the injected current, but still depends on the combination of the three biophysical parameters (which are assumed to be spatially uniform within the cell) in a non-trivial way due to the complex morphology of the neuron. In the absence of spikes, we used the mean squared difference between the simulated and the experimentally recorded voltage traces as the only error function, and restricted our attention to single-objective algorithms.

This benchmark proved to be the easiest in our entire test suite. Many algorithms found the best possible fit to the data in (almost) all the runs (Fig 3C; see also S4 Fig for a direct comparison between the best results provided by CMAES and random search), and most of them also converged relatively rapidly (Fig 3B and 3D). In this case, local search methods such as the Nelder-Mead and the L-BFGS-B algorithms also found the optimal solution efficiently in most runs. One curious exception was the DE algorithm implemented by the Inspyred package, which achieved a worse result than Random Search, even though the other implementation of the same algorithm by the Pygmo package was among the high-performing methods.

**Use Case 4: Simplified active model.**   This benchmark problem is more complex than the previous ones in several respects. The task in this use case is to determine the somatic densities of nine voltage-gated conductances in a model of a hippocampal CA1 pyramidal neuron with simplified morphology (consisting of only six compartments) so that the somatic voltage response of the model best approximates the response of a fully detailed CA1 pyramidal cell model under the same conditions (Fig 4). We used six of the error functions implemented by Neuroptimus (mean squared error excluding spikes, spike count, latency to first spike, action potential amplitude, action potential width, and after-hyperpolarization depth) to compare the two voltage traces. This also enabled us to test multi-objective algorithms besides the single-objective ones.

In this more complex use case, there were large differences in performance among the algorithms, with two orders of magnitude difference between the final errors of the best- and the worst-performing methods (Fig 4C; see also S5 Fig and S1 Table for a direct comparison between the best results provided by CMAES and random search at the level of individual voltage traces and error components). Once again, implementations of the CMAES algorithm achieved the best final scores, but the Pygmo implementations of PSO also delivered good final scores along with the best convergence speed. Among multi-objective algorithms, IBEA achieved the best final scores, and also performed quite well in terms of convergence speed. At the other extreme, all local search algorithms typically performed worse than Random Search, and are clearly inadequate for this type of problem. It is worth noting that all three implementations of the NSGA2 algorithm gave similar results, as did the different flavors of DE, although neither these algorithms nor several other bio-inspired algorithms (such as other evolutionary algorithms or ant colony optimization) were capable of providing as good solutions as CMAES and PSO on this benchmark.

**Use Case 5: Extended integrate-and-fire model.**   This use case involves fitting the parameters of an adaptive exponential integrate-and-fire model neuron so that it captures the spiking responses of a hippocampal CA3 pyramidal neuron recorded in vitro (Fig 5). This is a single-compartment model that does not include detailed models of neuronal biophysics; instead, it aims to capture neuronal spiking phenomenologically, using an extended integrate-and-fire formalism with an exponential term in the current-voltage relationship and an adaptation variable that is also linked to spiking [37, 38]. This model has a total of 10 parameters that had to

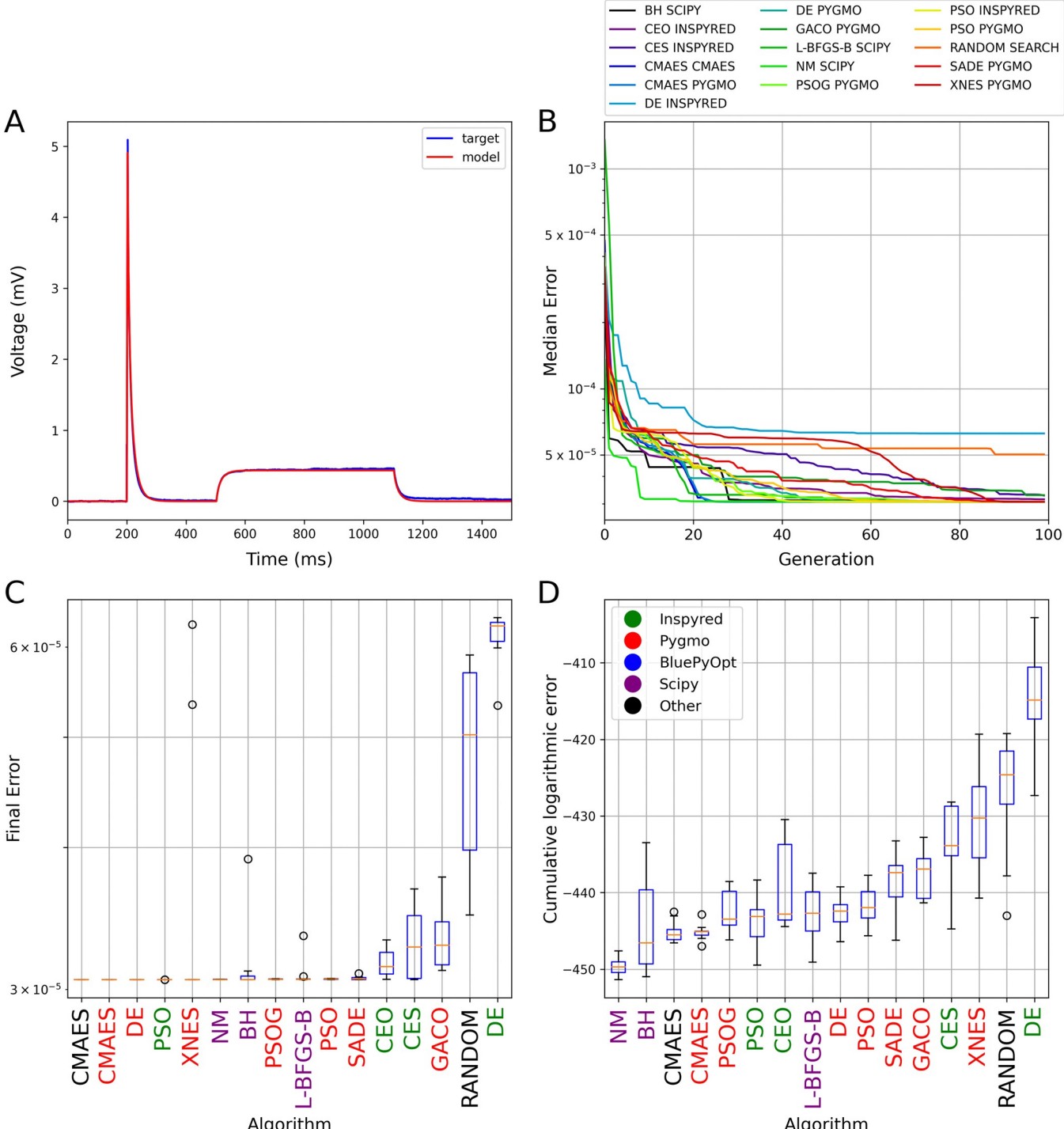

**Fig 3. The results of fitting the passive biophysical parameters of a morphologically detailed multi-compartmental model to experimental recordings from a hippocampal pyramidal neuron.** The plots in all four panels are analogous to those in Fig 1. Only single-objective methods were tested in this use case because only a single error function (mean squared difference) was used to compare model outputs to the target data. Panel A shows the results of a best-fitting model found by the CMAES algorithm.

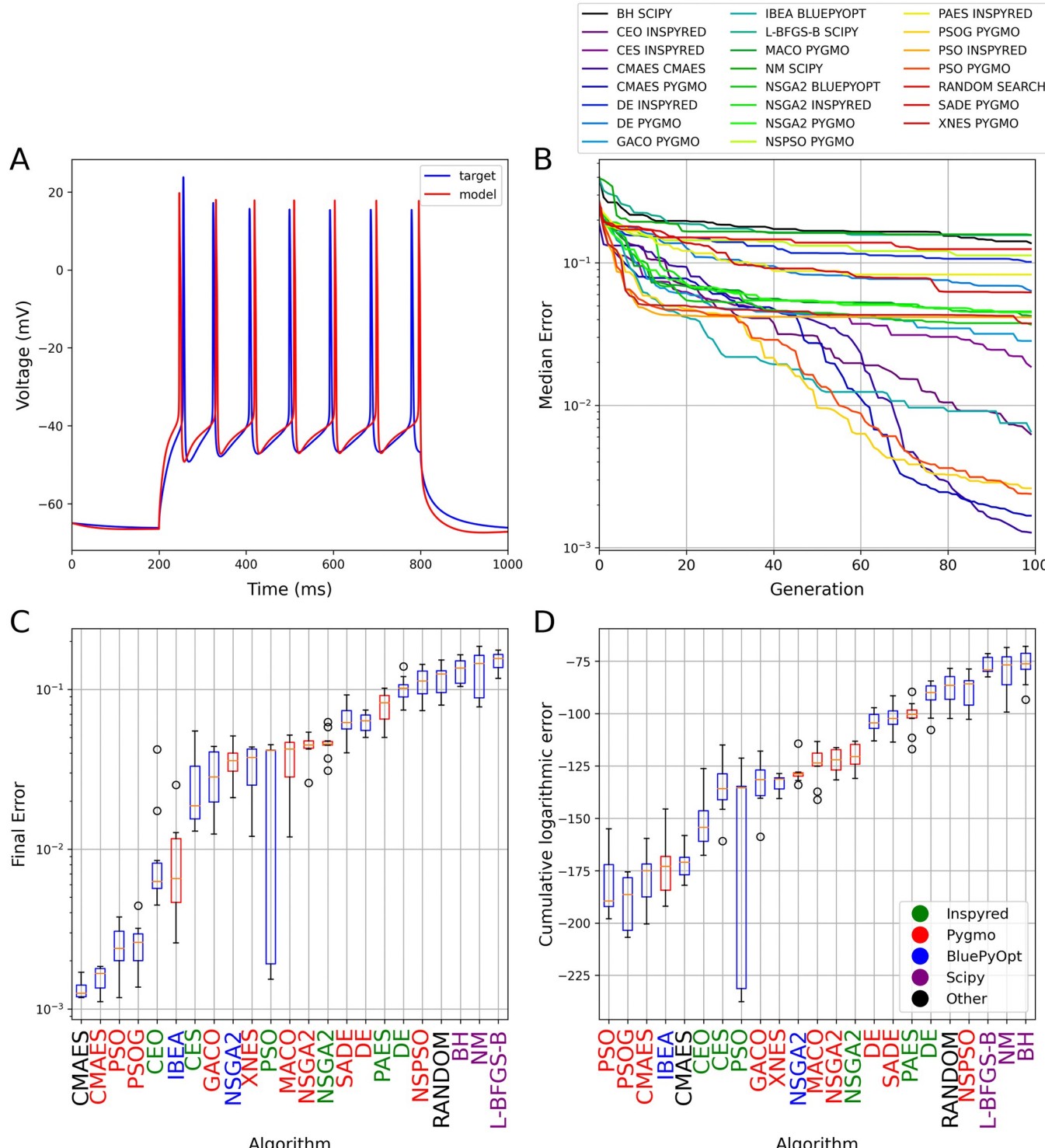

**Fig 4. The results of fitting the densities of somatic voltage-gated conductances in a morphologically simplified six-compartment model using a simulated voltage trace from a detailed compartmental model as the target.** The plots in all four panels are analogous to those in Fig 1. Panel A shows the results of a best-fitting model found by the CMAES algorithm.

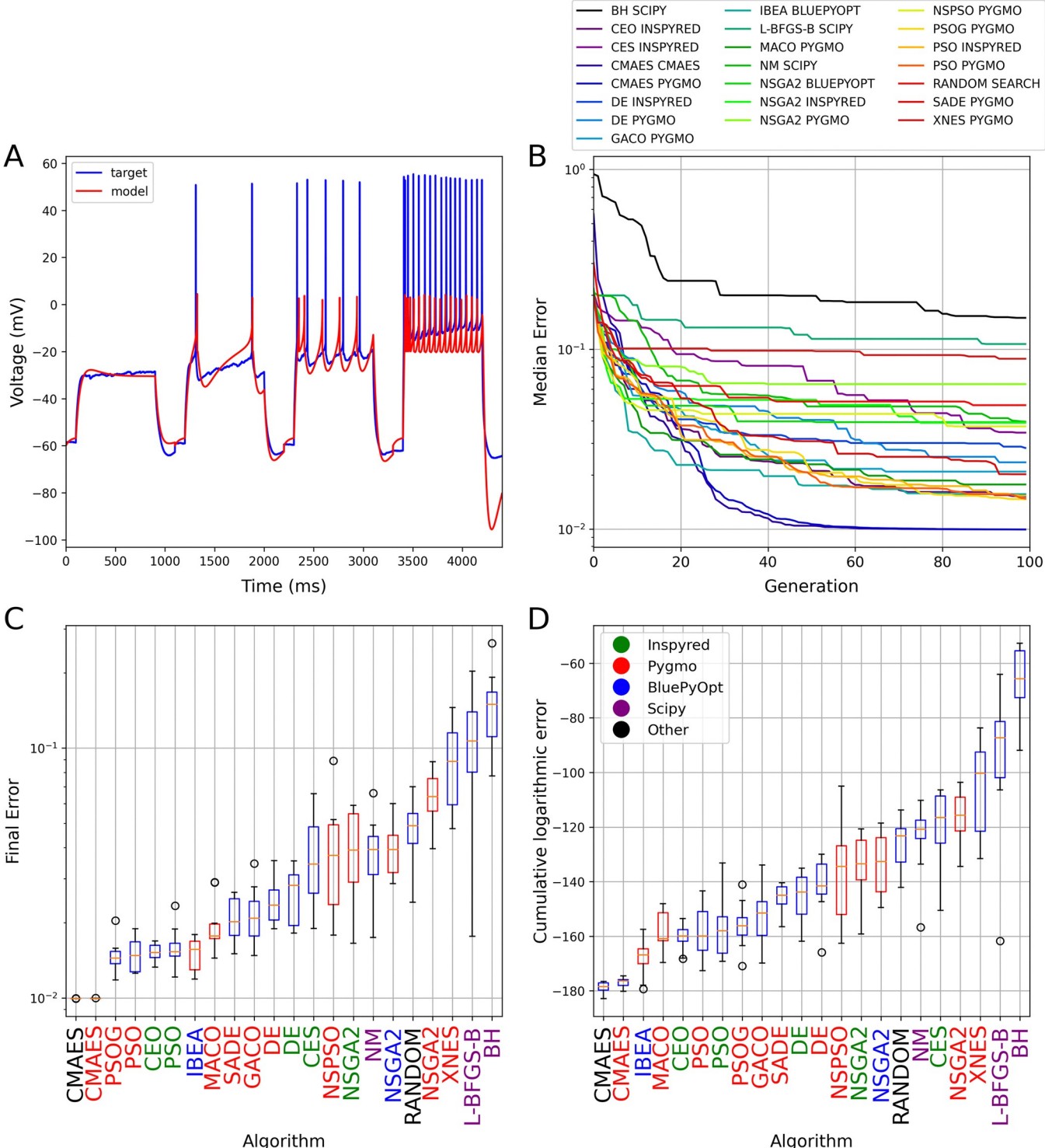

**Fig 5. The results of fitting a phenomenological spiking neuronal model (the adaptive exponential integrate-and-fire model) to capture experimental recordings with multiple traces.** The plots in all four panels are analogous to those in Fig 1. Panel A shows the results of a best-fitting model found by the CMAES algorithm. Note that the height of action potentials is irrelevant in the integrate-and-fire model, and the spikes generated by the model are not explicitly represented in the figure.

be fitted by the optimization algorithms. Unlike the models in the other use cases (which were implemented in NEURON), this model was implemented in the NEST simulator [39], and was treated as a black box by Neuroptimus. The parameters generated by the optimization algorithms were passed to an external Python script, which constructed the model, ran the simulations using NEST, and passed the results (spike times and subthreshold voltage traces in two separate files) back to Neuroptimus for evaluation and comparison with the experimental data. The data included the voltage responses of a real CA3 pyramidal cell to current steps of four different amplitudes (these responses are shown concatenated in blue in Fig 5A), and the model had to capture all of these responses simultaneously. As integrate-and-fire models cannot (and are not expected to) reproduce spike shape, we used spike count, latency to first spike, and the mean squared difference of the voltage excluding spikes as three error components during the optimization.

On this benchmark, the two implementations of CMAES found the solutions with the lowest error. In fact, they obtained the same lowest error score several times, and this was lower than the scores achieved by any other algorithm, so this error score likely corresponds to the best possible solution of this optimization problem. Although clearly inferior to CMAES on this problem, the various implementations of the particle swarm algorithm, the multi-objective algorithm IBEA, and the classical evolutionary algorithm found relatively good solutions, while several methods performed substantially worse than Random Search (Fig 5C; see also S6 Fig for a direct comparison between the best results provided by CMAES and random search at the level of individual voltage traces). We note that the PAES algorithm generated parameter combinations that led to errors during the NEST simulation, and was therefore excluded from the current comparison.

**Use Case 6: Morphologically and biophysically detailed CA1 pyramidal cell model.** Our next use case represents a typical scenario in the construction of morphologically and biophysically detailed compartmental models [1,4,7,40–43]. The model is based on the reconstructed morphology of a CA1 pyramidal neuron [44], and contains a large set of voltage-gated conductances, several of which are distributed non-uniformly within the cell (see Methods for further details of the model). The goal is to find the values of 12 parameters that determine the densities and biophysical properties of voltage-gated and leak conductances in the model such that the features extracted from the voltage responses of the model to multiple step current injections best approximate the average of the same features extracted from experimental recordings under matching conditions (Fig 6). One hyperpolarizing and five depolarizing current steps were used, and these yielded a total of 66 features of 20 different types (S3 Table) that were extracted and evaluated for each model instance during the parameter search. Examples of the experimental traces are given in S7 Fig.

Although this is certainly the most complex model in our benchmarking suite with the largest number of free parameters, finding solutions with errors close to the smallest possible value was apparently easier than in the previous two use cases (although, strictly speaking, we cannot rule out the possibility that none of the algorithms tested ever came close to the unknown globally optimum error score). More specifically, all three versions of PSO, both implementations of CMAES, and also the GACO and CEO algorithms consistently yielded similar low error scores, but several other algorithms, including the multi-objective IBEA and NSGA2 methods, also gave acceptable solutions. We note that running the PAES algorithm resulted in memory errors, and it was therefore omitted from the evaluation of this use case. It is also interesting to note that the actual feature values, and thus also the error components, showed considerable diversity among runs even for the best-performing algorithms (such as CMAES), although the diversity was much larger in the case of worse-performing ones (such as Random Search; S8 Fig).

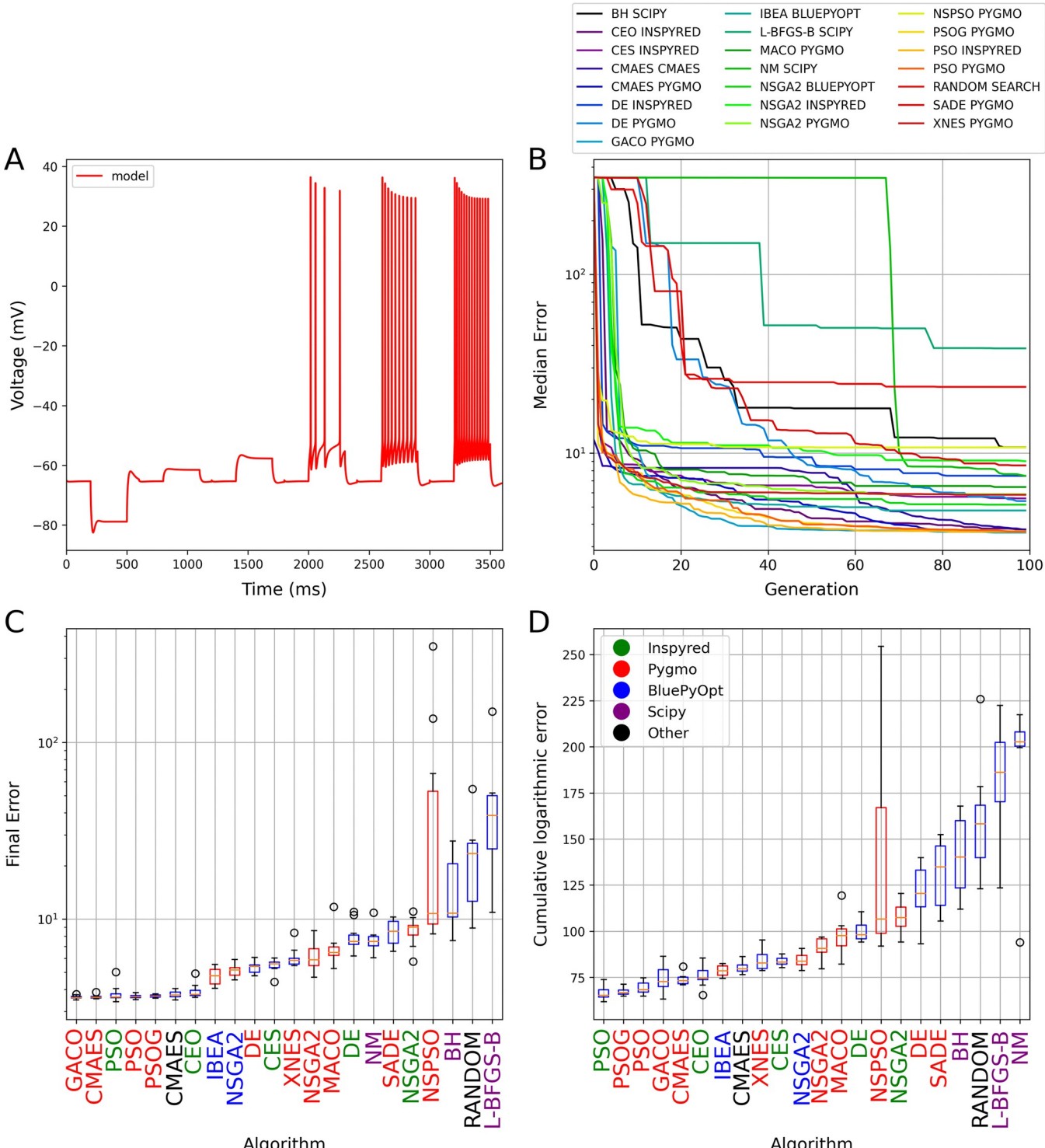

**Fig 6. The results of fitting conductance densities and kinetic parameters in a detailed CA1 pyramidal cell model.** The plots in all four panels are analogous to those in Fig 1. Panel A shows the results of a best-fitting model found by the CMAES algorithm. No target trace is shown because, in this use case, the actual target is defined by the statistics of electrophysiological features that are extracted from a set of experimental recordings.

## Overall performance of the algorithms

In general, no single algorithm is expected to perform well in all types of global optimization problems. Popular methods can take advantage of different types of regularities in the error function to speed up the search for the global optimum even in high-dimensional spaces with multiple local optima. Therefore, problems with different structures may require different algorithms for their efficient solution, and we can identify some signs of this heterogeneity when comparing the results of the individual benchmarks described above. Nevertheless, some clear patterns are evident, and we can quantify this by constructing and examining summary statistics for the algorithms across all the use cases.

Fig 7 summarizes the rankings of the various algorithms in our study according to final score and convergence speed. Individual ranks were based on the medians of the respective performance measure across all runs of the algorithm in a particular benchmark (this was also the basis of the placement of the algorithms along the horizontal axes in panels C and D of Figs 1–6), and Fig 7 shows the statistics of these ranks for each algorithm across the different benchmarks. Since multi-objective algorithms were tested only on the four problems that involved multiple error components, while single-objective methods were evaluated on all six use cases, the summary statistics in Fig 7 are based only on the four problems with multiple error components to avoid any bias. For comparison, S9 Fig shows the same statistics compiled across all six use cases, but only for single-objective methods.

In terms of the generally best-performing algorithms on our neuronal optimization test suite, the results are quite clear. In almost all cases, CMAES delivered the best results after 10,000 model evaluations, and its two implementations by different packages performed quite similarly. The three implementations of the particle swarm algorithm that we tested also showed similar performance, and were typically better than all the other methods except for CMAES. IBEA was close behind the PSO variants in the rankings, and was clearly the best among the multi-objective methods that we tested. It is interesting to note that some of the algorithms, including local search methods (and especially the Nelder-Mead algorithm) but also some other methods such as GACO and XNES showed widely varying performance across the different benchmarks, so these may be suitable for some problems but completely inadequate for others. Finally, the rankings based on the convergence score are generally quite similar to those based on just the final score, although there are some minor differences—for instance, PSO appears to be more competitive with CMAES according to this measure.

## Beyond single-cell biophysical modeling: optimizing a model of biochemical pathways related to long-term synaptic plasticity

All of the use cases we described so far concerned the electrophysiological behavior of single neurons. However, the use of Neuroptimus is not restricted to this domain; in fact, the software can be used to tune the parameters of any kind of model based on (essentially) any type of quantitative data. To demonstrate this versatility, in our final use case we applied Neuroptimus to optimize the parameters of a model describing the main subcellular biochemical cascades that shape postsynaptic plasticity in a single dendritic spine head (Fig 8A). We used a detailed, mass-action law-based model from [45] and handled it as a black box in Neuroptimus. The single-compartment biochemical model is implemented in Python and simulated with the reaction-diffusion (rxd) submodule of the NEURON simulator [46].

At the core of the model are three intracellular pathways: the calcium/calmodulin-dependent kinase II (CaMKII), the protein kinase A (PKA), and the protein kinase C (PKC) pathways, which are activated by four different input fluxes (calcium, glutamate, acetylcholine and β-adrenergic ligand). The activated cascades result in altered total synaptic conductance of

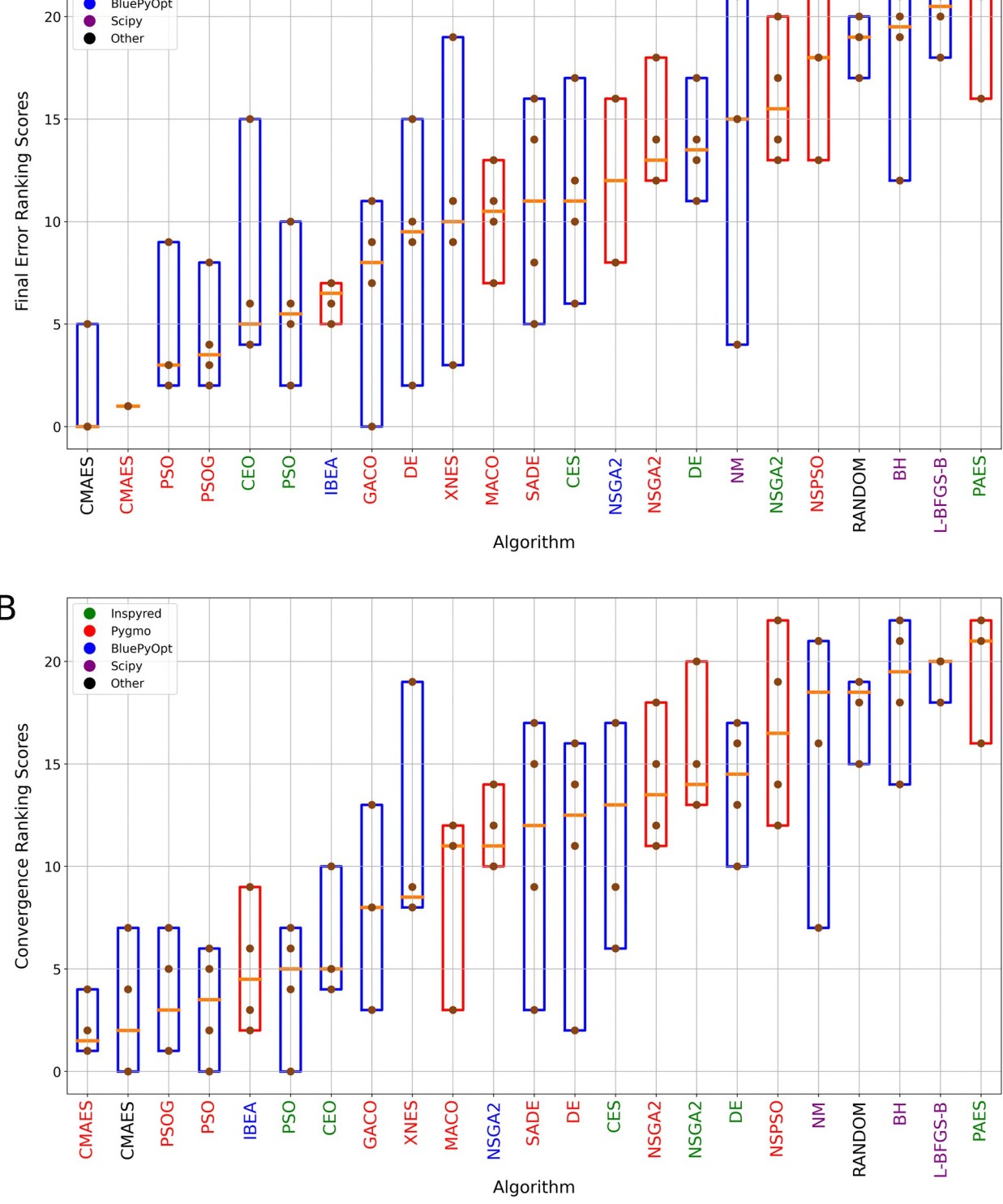

**Fig 7. Overall rankings of optimization algorithms.** Statistics of the ranks achieved by individual optimization algorithms on the different benchmarks involving multiple error components (Figs 1, 4, 5 and 6) according to the final error (A) and convergence speed (B). Brown dots represent the ranks achieved by the algorithms in each use case; boxes indicate the full range and the orange line represents the median of these ranks. The single-objective algorithms are shown in blue and the multi-objective ones in red boxes. The color of the name of the algorithm indicates the implementing package, with the color code included in the legend. Algorithms are sorted according to the median of their ranks.

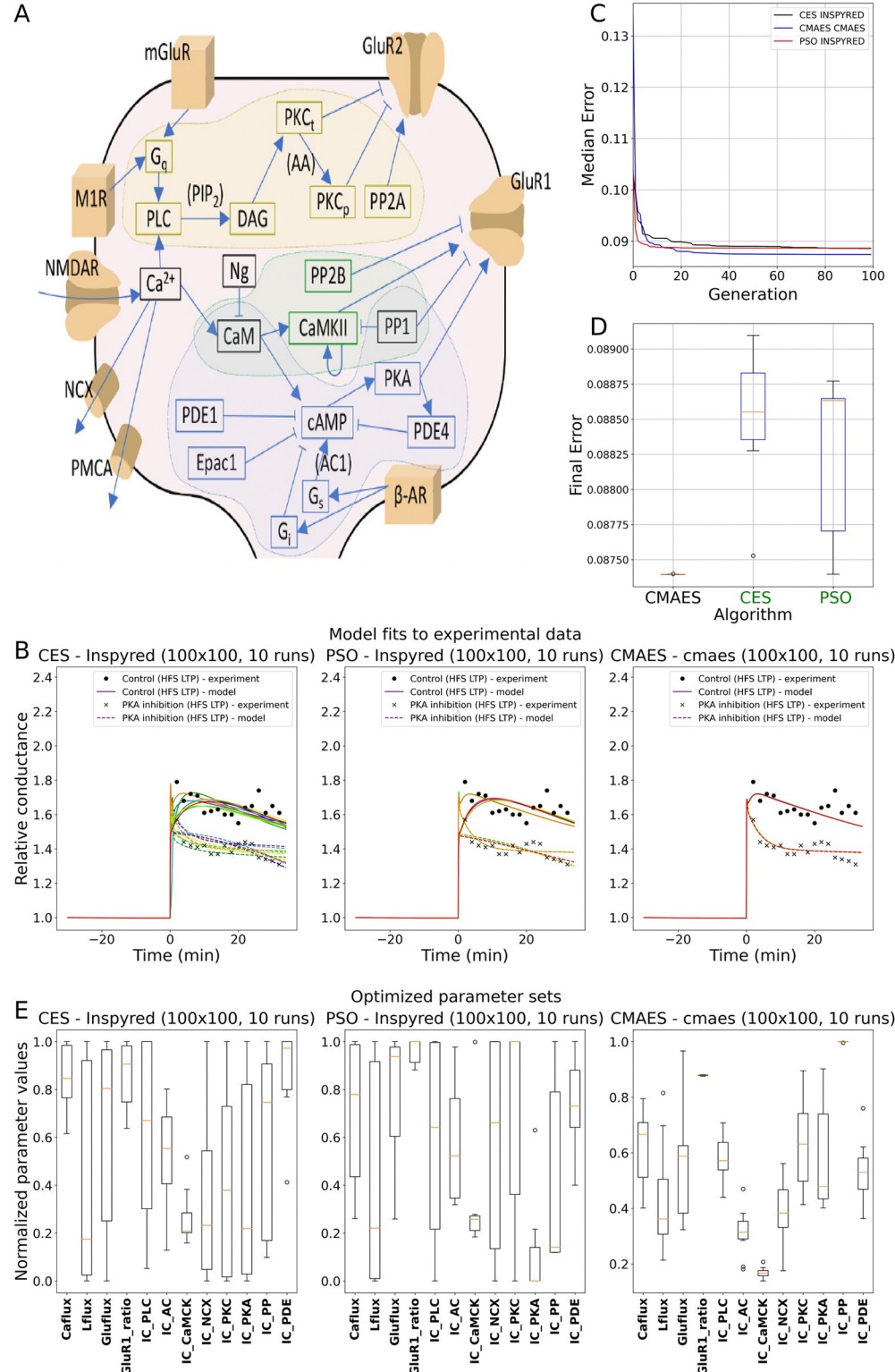

**Fig 8. The results of fitting input fluxes and initial concentrations of key molecular species in a subcellular biochemical network model.** (A) Schematic illustration of the intracellular signaling pathways included in the model (adapted from [45]). (B) Results of 10 model fittings to experimental data using the CES (Inspyred), PSO (Inspyred), and CMAES (Cmaes) algorithms. (C) Plot showing the evolution of the cumulative minimum error during the optimization. (D) Box plot representing the distribution of the final error scores over 10 independent runs of each

algorithm. (E) Box plot representing the distribution of the optimized parameters over 10 independent runs of each algorithm. A detailed description of the parameters can be found in the corresponding use case description in the Methods section.

AMPA receptors, which is often used as a measure of synaptic efficacy. The optimized parameters (12 altogether) include the input fluxes and the initial concentrations of the main molecular species, which are handled in functional groups. A detailed description of the model can be found in the Methods section, as well as in [45].

We selected two experimental data sets from [47] that were also included in the study that introduced the model [45], and used them as the target data to fit the model by optimizing its parameters. The data included two time courses of peak EPSP responses to synaptic stimulation before and after the induction of long-term potentiation. These were compared with the resulting time courses of total AMPA receptor-mediated synaptic conductance of the model using mean squared error as the cost function. Capturing long-term synaptic changes required simulating the model for several hours, and the time resolution had to be high enough to describe even the fastest chemical reactions in the system. Therefore, the simulation of the model with a single parameter set lasted for several minutes, and meaningful optimizations required parallelized algorithms and computational resources that could fully take advantage of these parallelization capabilities. Therefore, we ran this use case on the Neuroscience Gateway (NSG) using only three different algorithms (CES (Inspyred), PSO (Inspyred), and CMAES (Cmaes)). We ran each of these algorithms for 100 generations with a population size of 100, and repeated the optimization 10 times with different seeds.

The resulting fits are illustrated in Fig 8B. All three algorithms found good solutions, and there was relatively little difference in convergence (Fig 8C) and in the final error achieved (Fig 8D). However, we observed a clear difference in the variability of the solutions found. This is already visible in the variability of the final score; however, the difference in variability is even more striking when we look at the optimized parameters (Fig 8E). Of the three algorithms tested, the CES algorithm showed the greatest divergence, while the CMAES algorithm found solutions with almost identical errors, some well-restricted parameters and some other parameters with lower diversity than that of the other algorithms. Overall, these results clearly demonstrate that Neuroptimus offers a viable solution for the optimization of the parameters of biochemical networks, and, more generally, for parameter optimization of a wide range of models using a variety of different algorithms and computing platforms.

## Online database of optimization results

The results presented so far summarize the performance of a selected subset of the algorithms implemented by five Python packages, using their default settings, on a pre-defined suite of seven neuronal optimization problems. To increase the utility and reproducibility of our results, we also wanted to share the details of all the optimization runs, including the settings that enable their replication as well as their detailed results. In addition, we wanted to find a simple way of updating and extending the study with more optimization runs, potentially with different settings or algorithms not included in the present comparison, or even involving additional use cases, not just by us but also other interested researchers. We therefore designed, implemented and deployed an online database with an associated, publicly accessible web server (https://neuroptimus.koki.hu) that allows users to upload, query, and analyze optimization runs performed by the Neuroptimus software tool.

The website allows users to browse the optimization results stored in the online database, and filtering options are available to create lists of relevant results. The results of optimization

runs can be viewed in a detailed text-based format, and selected subsets of optimizations can be analyzed and displayed graphically, similarly to the plots in Figs 1–6 above. Registered users can also add to the database their own optimization results by uploading the JSON file (metadata.json) generated by Neuroptimus after each optimization run. Users can optionally also upload the other files that belong to the optimization (including the model and the target data) in the form of an archive, which creates an online record of the optimization that allows its full replication.

The database currently contains the results of all the optimization runs from the current study. This enables users of the website to replicate most of the figures in this paper, and to download individual optimization runs (including their settings and results). Users can also carry out custom analyses of the results, and (after registration) they can add their own optimization results (created using Neuroptimus) on both existing and novel use cases and compare these with other results on the same use case. This way, the website offers an interactive, continuously updated, and publicly accessible "live" version of this paper, which will provide a valuable online resource for researchers to explore and share methods and results on neuronal optimization.

## Discussion

The results of our study have the potential to advance the state of the art in neural parameter optimization in several different ways. First, we have created and shared Neuroptimus, a software tool that was designed to help both computational and experimental neuroscientists in the complete formulation and solution of neuronal parameter search problems. All the functions of Neuroptimus are accessible through a graphical user interface, although there is also a command line interface to support more advanced usage. Users of Neuroptimus gain uniform access to a large number of optimization algorithms implemented by several widely used Python packages, including several algorithms that were used successfully in previous neuronal modeling studies, and also several other state-of-the-art optimization methods that are popular in other domains but have not been applied to neuronal parameter optimization. This feature of Neuroptimus allowed us to systematically test the performance of a wide variety of parameter search methods on six distinct neuronal optimization problems, which makes it possible to offer some recommendations for future neuroscientific studies that rely on parameter optimization (see below). Finally, we created an online database of optimization results obtained by using Neuroptimus. This database currently contains the results of the present study, but the online user interface also allows us as well as others to add new results and compare them with already existing ones.

### Comparison of Neuroptimus with other neural optimization tools

A variety of software tools have been developed and used for the purpose of optimizing the parameters of neural models. This includes the built-in optimization modules of the general-purpose neural simulators NEURON [24] and GENESIS [25], the optimization-oriented features of the NetPyNE neural modeling framework [30], the Neurofitter program [26], as well as the Python packages BluePyOpt [27], NeuroTune [29], and pypet [28]. However, each of these tools (except for Neurofitter, which is no longer actively maintained, and focuses on a single specific cost function) relies on just one or a few algorithms, or a single external optimization package, to perform parameter search, while Neuroptimus provides access to a large variety of different algorithms from five distinct Python packages. This gives users of Neuroptimus a lot of flexibility to choose the best method for any particular fitting problem. As an example, Neuroptimus can take advantage of the large number of local and global

optimization algorithms offered by the Pygmo package [48], which is a mature and actively maintained tool used, among others, by the European Space Agency. Another distinguishing feature of Neuroptimus is its graphical user interface. Among the other tools, only NEURON and NetPyNE offer GUI-based access to parameter optimization, but the utility of this feature is limited by the availability of only a single basic optimization algorithm (the local search method PRAXIS in NEURON and grid search in the NetPyNE GUI).

## Algorithm recommendations based on our benchmarking results

The performance of optimization algorithms in general depends quite heavily on the nature of the problem, and no particular algorithm is expected to provide good solutions universally. However, within the task domain that we considered here, i.e., finding the biophysical parameters of models of single neurons, we can make some clear recommendations. Our benchmarking results were dominated by two different metaheuristics, covariance matrix adaptation evolution strategy (CMAES) and particle swarm optimization (PSO), followed by the multi-objective indicator-based evolutionary algorithm (IBEA)—so we would suggest trying these methods first to attack a novel neuronal parameter optimization problem. We also confirmed that local optimization algorithms are generally not suitable for more complex parameter search tasks, although they can be adequate and even efficient in the simplest cases. Finally, it is interesting to note that ant colony optimization (GACO) provided the best final scores for our most complex neuronal model (the detailed multicompartmental model of a CA1 pyramidal neuron), but it was not among the best performers on several simpler problems. In fact, GACO generally performed better on the more difficult benchmarks (compared to other algorithms), which may indicate that it is good at finding regions of the parameter space with low errors, but less effective in locating the exact location of the optimum within these regions. This is in contrast to algorithms such as CMAES, which appeared to be good at both finding the most promising regions of the parameter space and at zooming in on the precise location of the global minimum at later stages of the optimization.

Although, in principle, implementation details (particularly the default settings of algorithm parameters) could influence the performance of the algorithms, we found essentially no difference in the quality of solutions found by implementations of the same algorithms by different packages. However, the algorithms and even implementations of the same algorithm differ in the extent to which their execution can be parallelized, and this can have a large impact on the runtime of the algorithms, especially on highly parallel architectures. All algorithms of Inspyred and BluePyOpt, and some algorithms of Pygmo support the parallel evaluation of multiple candidate models (typically those within a particular generation or iteration), and Neuroptimus allows users to take advantage of these capabilities. It is worth noting in this context that Pygmo contains two variants of particle swarm optimization: PSO, which is closer to the original formulation of this algorithm, updates the velocities and positions of particles in a serial manner, and is thus not suitable for parallelization; and the generational variant PSOG, where the velocities and positions of all particles are updated in parallel. PSOG shares this feature with the implementation of PSO by the Inspyred package, and both of these can be run in a parallelized fashion from Neuroptimus. As a result, although all three variants of PSO produced similar final results in our tests, the runtimes of Pygmo's PSOG and Inspyred's PSO were significantly lower than those of Pygmo's PSO when multiple CPU cores were utilized. The situation is similar for the CMAES algorithm, where the current implementation in Pygmo does not support parallel evaluations. This was the reason for including the Cmaes package in Neuroptimus: this module implements CMAES in a way that allows straightforward parallelization, and Neuroptimus uses this implementation to support parallel execution

of this popular and efficient method. Finally, to demonstrate the importance of parallelized implementations, we note that running a single optimization of our most complex use case (the detailed CA1 pyramidal neuron) with 10,000 model evaluations took approximately 10 days on our compute server for algorithms without parallelization; by contrast, a single run of this use case using the same number of model evaluations (e.g., 100 generations with populations of 100 individuals) with algorithms that support parallel evaluations took only a few hours on a single node of a supercomputer (accessed via the Neuroscience Gateway) that allowed an entire generation of models to be evaluated in parallel.

## Comparisons with earlier studies

Our findings regarding the relative performance of various optimization methods are mostly in line with results of earlier studies that included such comparisons. Vanier and Bower [21] compared four different algorithms on a set of use cases similar to ours. They examined the performance of random search, conjugate gradient descent (a local search method), an evolutionary (or genetic) algorithm, and simulated annealing. They found that their evolutionary algorithm (which was similar to the CEO algorithm in our benchmark) delivered good performance even for more complex use cases with a larger number of parameters. This is consistent with the generally good results of evolutionary-type algorithms in our study, although we found several more recent variants that outperformed the classic version. They also found simulated annealing to be very effective, and this was later confirmed by Friedrich et al. [15] using the implementation that is built into the GENESIS simulator. None of the packages currently supported by Neuroptimus contains the traditional simulated annealing algorithm. In fact, older versions of the SciPy module used to include simulated annealing, but it was later deprecated and replaced by the basinhopping algorithm, which is considered to be a generalization of simulated annealing. In this light, the generally poor performance of the basinhopping algorithm in our tests is slightly surprising, although it may be caused by implementational problems or improper default settings of the parameters.

Our finding that CMAES performs well in a variety of different tasks is supported by several other studies. In particular, CMAES and IBEA have been compared on data-driven neuronal models, and CMAES generally delivered better final scores [49]. CMAES was also found to be efficient and robust in a study that involved fitting the biophysical parameters of models of striatal neurons [50]. Outside the neuronal modeling domain, a recent study compared two sophisticated evolution strategy variants, CMAES and xNES on different problems, and the results clearly showed that CMAES consistently outperformed xNES [51]. Our findings also support this conclusion, and add some evidence regarding two additional evolution strategy types: classic evolution strategy (CES, from the Inspyred package), which performed similarly to xNES (from Pygmo), and Pareto-archived evolution strategy (PAES, from Inspyred), which was one of the weakest performers (worse than random search) in our comparison.

Finally, in our benchmarking results, the self-adaptive variant of Differential Evolution (SADE) performed quite similarly to the original algorithm (DE), which was in contrast to the results of an earlier comparative study that found SADE to be clearly superior in a variety of real-world optimization problems [52].

## Limitations of the benchmarking study

The specific results that we obtained in our benchmarking study depend, to some extent, on some arbitrary choices that we had to make when designing our tests. For instance, we arbitrarily set the number of model evaluations to 10,000 for every algorithm to ensure a fair comparison of the final results. However, not all algorithms converged completely after 10,000

model evaluations in some of our use cases, and thus allowing more (or fewer) evaluations would likely affect the rankings based on the final score. The other performance measure that we used, the convergence score, is expected to be less sensitive to the exact number of model evaluations allowed, and also provides an indication of the speed of convergence.

We made another arbitrary choice for every algorithm with a hierarchical design, which includes all population-based methods (such as evolutionary and swarm intelligence algorithms) but also other nested algorithms such as basinhopping. We set the number of model evaluations in the innermost loop (i.e., the size of the population, or the number of steps in the local optimization) to 100, and the number of iterations in the outer loop (e.g., the number of generations) also to 100. This 100x100 partitioning of the total of 10,000 evaluations is a reasonable choice for most algorithms (as it allows sufficient diversity within populations but also a sufficient number of iterations for convergence), and led to good results in most cases. This choice also allowed us to make near-optimal use of the computational resources offered by the Neuroscience Gateway. However, it is entirely possible that a different choice (such as 200 generations with a population size of 50, or the other way around) would have resulted in improved performance for some of the algorithms.

Similarly, almost all the algorithms that we used include some tunable parameters that change the course of the optimization and may heavily influence the quality of the results. Several algorithms also have multiple variants that differ in important algorithmic or implementational details; for instance, evolutionary algorithms are based on the cyclic application of operators such as selection, mutation, crossover and replacement. Each of these operators can have several variants (and parameters for each variant) that change the behavior of the algorithm and may heavily affect the quality of the solution.

We decided to use the default settings specified in the optimization packages for all algorithms (with the exception of the CEO algorithm from the Inspyred package—see the description of this algorithm in Methods for details). In many cases, these settings were compatible with parameters recommended in the relevant literature; in some other cases, different sources suggested different settings; and in some cases, no such recommendations could be found. It is also known that the best settings for such algorithmic parameters can depend on the details of the problem, so it is possible that distinct settings would be optimal for the various use cases. For example, it is known that the choice of hyperparameters, especially the step size and the social rate, can have a large effect on the convergence of Particle Swarm Optimization. Without relying on neighborhood communication the algorithm searches locally, converges faster, but may get stuck in local minima, especially in higher dimensions. At a higher social rate, PSO converges more slowly but has a higher chance of finding the global optimum [53]. Therefore, it is quite likely that the same default social rate is not optimal for all of our use cases with different numbers of tunable parameters, and the results could be improved on certain use cases by varying this hyperparameter.

On the other hand, we think it is valuable to know which algorithms and implementations provide consistently good results even without fine-tuning their hyperparameters, as this provides a reliable starting point for any parameter optimization study, and allows even non-expert researchers to apply these methods effectively in their projects.

Overall, we may conclude that the rankings that we provide are not just about the algorithms themselves (or even about their implementations by particular packages), but are likely also influenced by the settings of the parameters (including the population size) for each method. In fact, we expect that the results of some algorithms could be improved substantially by using different settings, and there are also many additional optimization algorithms that were not included in the current comparison. Therefore, we hope to update and extend our study as more optimization results become available (see below for further discussion).

Finally, we note that the two simple performance measures that we used may not capture some characteristics of the optimization results that users might be interested in. For instance, it could be useful to provide some measure of the diversity of the final population; in some contexts, such as the construction of neuronal populations for a network model, high diversity may be desirable, while in other cases, high diversity may indicate incomplete convergence of the optimization algorithm. Users may also be interested in the evolution of specific error components during the optimization. While it is certainly possible to construct diversity measures (and other performance metrics) and look at the detailed error scores from the data saved by Neuroptimus, we did not systematically evaluate them in the current study.

## Possible extensions

In addition to testing our use cases with more algorithms and settings as described above, the results of our study could be extended in other important ways. One possible direction would be to extend the set of use cases to other types of problems. Most of the use cases included in the current study involved single cell models of varying complexity. In the last use case, we showed how Neuroptimus can be used outside this domain, to tune the concentrations of molecules in biochemical pathways involved in synaptic plasticity, relying on the reaction-diffusion (rxd) module of NEURON to run the simulations. However, in principle, Neuroptimus can optimize the parameters of almost any type of model using many kinds of target data and error functions. One particularly exciting possibility is to tune the synaptic weights (or other parameters) in a network model to reproduce measures of population activity or to perform specific computations, which is an important emerging topic in computational neuroscience [18,19,20,54,55]. This can be implemented in Neuroptimus by using an appropriate external simulator (such as NEST or Brian) to simulate the network dynamics, and using the relevant activity measures (such as average firing rate or oscillation frequency) as the target data.

There are also many useful features that we could potentially add to our optimization tool, Neuroptimus. For example, Neuroptimus currently returns only a single parameter combination corresponding to the lowest error score at the end of the optimization (although the parameters and errors of all the models tested are also saved into a file and may be analyzed outside Neuroptimus). This is the case even when multi-objective methods are used; the winner in this case is selected from the final population by minimizing the weighted sum of the objectives, using weights provided by the user before the optimization run. However, the final population of multi-objective optimization carries much more information, as it approximates the Pareto front (the parameter combinations representing the best possible tradeoffs between the objectives for different choices of the weights). Therefore, it would be useful to add to Neuroptimus the capability of properly representing and analyzing the results of multi-objective optimization. Other useful extensions could include the ability to chain optimization algorithms (e.g., by automatically running local optimization using the output of a global algorithm as the starting point), and the ability to visualize the progress of optimization (in error space and in the parameter space) while it is still running.

A recent study [17] advocated using a two-step approach to fitting the parameters of complex multicompartmental models, whereby the subset of parameters that are sufficiently constrained by subthreshold physiological responses are tuned first using a local search algorithm, and these parameters then remain fixed while the rest of the parameters are tuned using both suprathreshold and subthreshold responses via an evolutionary algorithm. Such multi-step approaches can decrease the number of simultaneously optimized parameters, which makes it easier for the search algorithms to find good solutions, but require that particular parameters affect only certain types of behaviors, which may not always be the case. Neuroptimus

currently supports such multi-step approaches only indirectly, requiring manual modifications in the optimization setup between steps, but could potentially be extended to handle such workflows automatically.

### Community and cooperation through the Neuroptimus website

We do not see the benchmarking results presented in this paper as the final word in evaluating parameter search methods for neuroscientific problems. As we argued above, it will be important to extend our study with more use cases and further evaluation of different algorithms and settings. Global parameter optimization is also a fast-moving field where new methods emerge regularly; the fact that Neuroptimus supports several actively developed packages, and is also flexible enough to accommodate new packages guarantees that new developments can be integrated with minimal effort.

We have developed and deployed the Neuroptimus web server to provide a platform for sharing and analyzing optimization results. By allowing all users to upload results obtained by using Neuroptimus, and to compare them with already uploaded results (including all the results of the current paper), the web site will become a continuously updated "live" version of this paper. This should facilitate meaningful, quantitative comparisons of parameter optimization methods, and aid the collaboration of different research groups that are interested in this topic. We encourage all interested professionals (and especially those who are experts in using particular algorithms) to run the use cases with improved settings, try other algorithms, add new use cases, and share their results on the Neuroptimus website. This way, we can collectively track new developments, and offer reliable solutions for an increasing variety of neural optimization problems.

## Methods

### Software tools and services

**Neural Optimization User Interface (Neuroptimus).** At the core of our methodology is a software tool that we developed, called Neural Optimization User Interface (or Neuroptimus). Neuroptimus implements a software framework that allows users to set up and solve parameter optimization problems and analyze the results. Neuroptimus performs parameter optimization mainly by providing a common interface to a large number of popular parameter search algorithms implemented by various open source packages. In principle, Neuroptimus can be used to optimize the parameters of all kinds of systems; however, its main purpose is to aid parameter fitting in neural systems, and especially in detailed models of neurons. Accordingly, it includes many features that were developed specifically for this scenario, which support simulating biophysical models of neurons using the NEURON simulator, and comparing their behavior to experimental data obtained with common electrophysiological protocols.

Neuroptimus is essentially an updated and extended version of our previous tool Optimizer (https://github.com/KaliLab/optimizer) [15]. The basic design of these two pieces of software is quite similar, and they also share many details of their implementation. Therefore, we will focus on the new features and other differences here, and summarize the features that are used by the current benchmarking study, but we refer the reader to Friedrich et al. (2014) and the Neuroptimus documentation (https://neuroptimus.readthedocs.io/) for further details.

Neuroptimus is open source software, implemented in Python3, and can be accessed at the GitHub repository https://github.com/KaliLab/neuroptimus. Its functions are available both via a graphical user interface (GUI) that guides users through the steps of setting up, running, and evaluating the results of parameter optimization tasks, and via a command line interface that performs these tasks based on the settings stored in a configuration file. The GUI was

built using the PyQt5 package that provides a Python binding to the cross-platform GUI toolkit Qt (version 5).

The complete definition of a neural parameter optimization problem requires the specification of multiple components. First, we need to provide the model whose parameters we wish to optimize. Neuroptimus can load, manipulate and execute models implemented in the NEURON simulator (either in Python or its native HOC language). The parameters to be optimized can be selected from the parameters of this model, or the user can provide a function (implemented in Python) that defines abstract parameters and how these should be mapped onto the concrete parameters of the NEURON model. As an alternative, models can be implemented by any external program that is capable of reading the variable parameters of a model candidate from a text file, setting up the model accordingly, running the simulation(s), and saving the results to files that can be interpreted by Neuroptimus.

Second, the cost function for neural parameter optimization is typically defined in terms of some target data (from experiments or prior simulations) and a function (or set of functions) that quantifies the difference between the output of the model and the target data. Neuroptimus can handle different types of target data, including time series (such as voltage and current traces), explicit spike times, and feature statistics.

Neuroptimus implements several error functions that can be used individually or in combination to evaluate during the optimization process the discrepancy between the voltage traces (or other time series) generated by the optimized model and the target data [15]. These cost functions range from general ones such as the mean squared error to more specific ones that are useful mainly in the context of fitting neuronal voltage responses and characterize the pattern and shape of action potentials (Table 1).

The error functions above (which were already present in Optimizer; [15]) compare each voltage (or current) trace generated by a model with a specific voltage (or current) trace in the target data. However, a common task in single cell modeling involves finding model

**Table 1. Cost functions implemented in Neuroptimus.**

| Feature name | Definition |
|---|---|
| Mean squared error | Mean squared difference between the model trace and the target trace point by point, normalized by the squared range of the target data |
| Mean squared error (excluding spikes) | Same as above but excludes the parts of both traces in the vicinity of action potentials (in either trace) |
| Derivative difference | Normalized mean squared difference of the temporal derivatives of the two traces |
| Spike count | Absolute difference of the number of spikes in the entire traces, normalized by the sum of the two spike counts |
| Spike count (during stimulus) | Identical to spike count, except it only takes into account the action potentials during the stimulus |
| ISI difference | Sum of the absolute differences of the inter-spike intervals of the two traces, normalized by the length of the traces |
| Latency to 1st spike | Squared difference between the time to the first spike from the start of the stimulus in the two traces, normalized by the squared length of the traces |
| AP overshoot | First calculates the amplitudes of the action potentials in both traces as the difference between the AP peak voltage and the AP threshold, then takes the mean squared difference of the AP amplitudes normalized by the squared maximal amplitude of the target trace |
| AP width | Mean squared difference between the width of the action potentials at their base (at the threshold voltage level), normalized by the squared mean width of the APs in the target trace |
| AHP depth | The squared mean of the difference in the corresponding after-hyperpolarization depths, normalized by the squared range of subthreshold potential in the target trace |

parameters such that the behavior of the model becomes similar to the typical behavior within a set of experimentally recorded neurons [5, 7]. In this case, it is more natural to define the target of optimization as the mean values of a set of pre-selected features extracted from the experimental voltage traces (which may come from several experiments involving the same or different neurons). Then the natural way of defining error functions is by evaluating the difference between the value of a particular feature extracted from the voltage response of the model and the mean value of the same feature in the experiments, divided by the standard deviation of the feature in the experimental data. One additional advantage of this definition is that it provides standardized, dimensionless error scores that may be combined in a straightforward manner.

This approach based on feature statistics is now supported by Neuroptimus. To provide access to a diverse array of electrophysiological features, and ensure compatibility with some common workflows [5,7,54,56], Neuroptimus utilizes the Electrophys Feature Extraction Library (eFEL; https://github.com/BlueBrain/eFEL) [31] to characterize the voltage responses of the models. The target data in this case contain the experimental mean and standard deviation values of a predefined set of eFEL features extracted from voltage responses to specific current step inputs, stored in a JSON file created from the recordings using the BluePyEfe tool (https://github.com/BlueBrain/BluePyEfe) and a custom script that converts the output of BluePyEfe to the format expected by Neuroptimus. This JSON file also contains the full specification of the stimulation protocols. When the optimization is run using the GUI, the settings of the stimuli and the features are automatically loaded into the GUI from this input file. During the optimization process, in every model evaluation step, the features included in the input file (and selected in the GUI) are extracted from the model's voltage traces, and errors are computed for every feature using the feature statistics-based error function described above [56].

Recent studies classify optimization problems according to the cardinality of objectives as single, multi- (2–3 objectives) and many-objective tasks (more than 3 dimensions), which affects the nature and the complexity of the problem [57]. However, we characterized our problems simply as single- or multi-objective problems because these require different internal representations and are solved by different algorithms. Multi-objective problems involve several objective functions that are to be minimized simultaneously and require finding a set of solutions that give the best tradeoffs between the objectives.

Neuroptimus makes it possible to use arbitrary weighted sums of error functions as the ultimate objective function of the parameter search. When single-objective algorithms are used, the weighted sum is calculated for every model during the optimization process, and is used as the objective function. In the case of multi-objective algorithms, all the error functions are treated as separate objectives during the optimization, but the weighted sum is still used after running the search to select a single preferred solution from those returned by the algorithm [5,27].

Neuroptimus supports parameter optimization algorithms implemented by five external Python packages (Pygmo, Inspyred, BluePyOpt, SciPy, and Cmaes), and also contains an internal implementation of a simple random search algorithm that takes independent, uniformly distributed samples from the entire search space. Pygmo is a general-purpose scientific Python library for optimization, based on the C++ library pagmo, which implements many different optimization algorithms in a common framework [58]. Inspyred is a Python library specifically developed for bio-inspired (mainly evolutionary) computation, and was already supported by Optimizer [15]. The Blue Brain Python Optimization Library (BluePyOpt) is a software framework developed at the Swiss Blue Brain Project [27], which implements multi-objective optimization algorithms including the Indicator Based Evolutionary Algorithm (IBEA), and has

been applied successfully in several computational neuroscience projects [5,7,54,59–63]. SciPy [64] provides implementations of various methods for scientific computation, and includes several basic optimization algorithms, some of which were already supported by Optimizer [15]. Finally, we also included the Cmaes package because it provides a simple, robust, and easily parallelizable implementation of the Covariance Matrix Adaptation Evolution Strategy (CMAES) algorithm, a popular and powerful search method that is also included in Pygmo but in an implementation that does not support the parallel evaluation of models within a population.

Some of the algorithms are local (essentially gradient-based) search methods, but most of them are based on metaheuristics that attempt to find the global minimum of the cost function (s). Many of the most popular single- and multi-objective optimization algorithms are included. Most of the algorithms also have parameters that are configurable through the GUI or the configuration file.

Solving nontrivial parameter optimization problems typically requires the evaluation of many parameter combinations. In our case, this corresponds to running a large number of simulations, which may take a prohibitively long time if simulations are performed sequentially, especially for complex models such as morphologically detailed neurons, circuits, or multi-scale models that include biochemical or molecular processes. Fortunately, many global optimization methods (including evolutionary and swarm intelligence algorithms) can be implemented in a way that populations of models can be evaluated in parallel, and several such parallel (or easily parallelizable) implementations are included in the Python libraries supported by Neuroptimus. However, Python provides several different methods for parallel execution of code, and the optimization packages we use differ in terms of which parallelization approaches they support. As a consequence, Neuroptimus uses the *multiprocessing* module for the parallel execution of algorithms in Pygmo, Inspyred and Cmaes, while it relies on the IPython Parallel (*ipyparallel*) package to run the algorithms of BluePyOpt in parallel. We note that some optimization algorithms cannot be efficiently parallelized, while for some others (including several in the Pygmo package) parallel execution is not currently supported by the optimization library.

Batch evaluation of the models is a requisite to use one of the various parallelization strategies in Neuroptimus. Therefore, both internal and external evaluations have to generate results simultaneously. If we used a single model instance in every process, the results could be mixed or swapped. Therefore, when simulations are carried out within Neuroptimus (using NEURON), a new model instance is created for every parameter set generated by the selected algorithm, and every evaluation running in parallel is performed with a separate model. In case of using the external simulator, every individual is evaluated in a separate subprocess, and files with unique names are used for communication between Neuroptimus and the external simulation script.

We list all of the available algorithms along with their basic properties in Table 2. Many of these algorithms were tested in our benchmarking study, and these will be described in more detail below.

The easiest way to perform parameter optimization using Neuroptimus is by using the GUI, whose seven tabs guide the user through the steps of setting up, running, and evaluating the results of the parameter search. The GUI allows the user to load the target data, select the model and the parameters to be optimized, set up the stimulation and recording conditions, configure the error function(s), run the parameter search, and then visualize and analyze the results. The final as well as intermediate results of the optimization are also saved to files, and can be analyzed outside Neuroptimus. This includes the parameters and errors of each simulated model as well as the statistics of generations saved into text files, the voltage trace of the

**Table 2. Algorithms included in Neuroptimus.** The properties listed include the full name of the algorithm, the abbreviation used in this article, the type according to the number of objectives (single/multi-objective), the implementing package(s), and the method of parallelization used in Neuroptimus (None if only serial execution is supported).

| Algorithm | Objectives | Packages | Parallelization |
|---|---|---|---|
| Custom Evolutionary Optimization (CEO) | Single | Inspyred | multiprocessing |
| Classic Evolution Strategy (CES) | Single | Inspyred | multiprocessing |
| Particle Swarm Optimization (PSO) | Single | Inspyred | multiprocessing |
| | | Pygmo | None |
| Non-dominated Sorting Genetic Algorithm (NSGAII) | Multi | Inspyred | multiprocessing |
| | | Pygmo | |
| | | Bluepyopt | ipyparallel |
| Differential Evolution (DE) | Single | Inspyred | multiprocessing |
| | | Pygmo | None |
| Pareto Archived Evolution Strategy (PAES) | Multi | Inspyred | multiprocessing |
| Basin-Hopping (BH) | Single | SciPy | None |
| Nelder-Mead (NM) | Single | SciPy | None |
| limited-memory Broyden-Fletcher-Goldfarb-Shanno algorithm with bound constraints (L-BFGS-B) | Single | SciPy | None |
| Self-Adaptive Differential Evolution (SADE) | Single | Pygmo | None |
| Covariance Matrix Adaptation Evolutionary Strategy (CMAES) | Single | Pygmo | None |
| | | Cmaes | multiprocessing |
| Exponential Natural Evolution Strategies (XNES) | Single | Pygmo | None |
| Extended Ant Colony Optimization (GACO) | Single | Pygmo | multiprocessing |
| Multi-objective Hypervolume-based Ant Colony Optimization (MACO) | Multi | Pygmo | multiprocessing |
| Particle Swarm Optimization Generational (PSOG) | Single | Pygmo | multiprocessing |
| Non-dominated Sorting Particle Swarm Optimization (NSPSO) | Multi | Pygmo | multiprocessing |
| Indicator Based Evolutionary Algorithm (IBEA) | Multi | Bluepyopt | ipyparallel |
| Simulated Annealing (SA) | Single | Inspyred | multiprocessing |
| Praxis | Single | Pygmo | None |
| Simple Genetic Algorithm (SGA) | Single | Inspyred | multiprocessing |
| Estimation of distribution algorithm (EDA) | Single | Inspyred | multiprocessing |
| Artificial Bee Colony (ABC) | Single | Pygmo | None |
| Differential Evolution 1220 | Single | Pygmo | None |

best model saved into text files and in several image formats, and a final summary of the optimization process and the results saved into an HTML file for visual inspection through a web browser and a JSON file for a machine-readable non-SQL data representation. This final metadata file created after the optimization contains automatically generated names for the optimization and the model, details of the parameters of the model (name, boundaries and optimal values), details of the error functions used to calculate the final error (name, value, weight, weighted value), settings of the target data, the algorithm and package used for the optimization, parameters used by the algorithm, and finally the statistics of each generation.

The program also saves the full configuration of the optimization task, and the resulting configuration file can be used (directly, or after suitable modifications) by the command-line interface of Neuroptimus to re-run the optimization (with the same or modified settings). This method was used in our benchmarking study to run batches of the same optimization with different random seeds, using a simple Python script to edit the configuration file and create multiple versions of the optimization task.

**Neuroptimus server.**    To share our results in a way that allows easy replication and further analysis, and to enable the straightforward extension and updating of the current study, we created an online database of optimization results that is accessible via a web interface. We designed, created and deployed the Neuroptimus web-server, which can be publicly accessed at https://neuroptimus.koki.hu and enables all users to browse, view and analyze the optimization results stored in the database. Furthermore, authenticated users can also upload their optimizations and compare their results with previously uploaded ones.

The Neuroptimus server structure consists of an Nginx web server that handles the requests and responses, the frontend implemented using the JavaScript library ReactJS, the backend created in the Python web framework Django, backed up by a PostgreSQL database connection. The site handles the authentication of registered users, the uploading of optimization results via a web form, the visualization of the data in a table structure, and the creation of plots for comparison. The database stores information about the optimization itself, the model used for the optimization and its parameters, the algorithm and its configuration, details about the target data, the statistics of each generation produced by the algorithm, the creation time of the results and the upload time. The metadata JSON file created by Neuroptimus can be uploaded to the server and all of its information content is transferred to the database automatically. Optionally the compressed optimization files can also be uploaded and subsequently downloaded. Analysis of the optimization can be created semi-automatically by selecting the desired algorithms for comparison and visualizing them on the charts. Thus far generation plots, final and convergence score box plots are available for online observation.

During the registration process users need to provide their name, affiliation, and email address, choose a username, and create a password. Verifying email addresses grants permission for users to upload their optimizations. Forgotten passwords can be reset on the website via email verification.

## Optimization algorithms

In the current study, we evaluated a large set of parameter search algorithms, including several of the most widely used single-objective and multi-objective methods. Our optimization tool supports optimization algorithms implemented by five separate Python packages: Inspyred [65], Pygmo [58], BluePyOpt [27], Cmaes [66], and SciPy [64]. Table 2 shows which packages implement each of the supported algorithms. The majority of these algorithms can be categorized as evolutionary or nature-inspired metaheuristics.

Due to constraints on time and computational resources, we could not include every single algorithm supported by Neuroptimus in the detailed comparison that we performed using our neural benchmarking suite (see below). However, we aimed to provide good coverage of algorithms that were used previously in neuronal optimization [9,15,21,27], and also included several additional algorithms that consistently provided good performance in other settings [67–70].

Finally, we added some basic search methods such as uniform random sampling and two widely used local optimization algorithms to provide a baseline against which we can measure the performance of more sophisticated methods. We provide a brief description of the algorithms tested in our neural optimization benchmark below; all the relevant hyperparameters and settings for each of these algorithms are provided in S2 Table.

**Baseline algorithm.**    The *Random Search (RAND)* algorithm is the simplest heuristic to discover solutions by trial and error. This is our baseline method, which samples parameters from the search space repeatedly based on the uniform probability distribution. Neuroptimus uses our own implementation of this method [71].

**Local optimization algorithms.** The *Nelder-Mead (NM)* algorithm is a classic simplex-based direct search method to find a local minimum of the cost function. It uses n+1 points to create a polygon (or simplex) in the n-dimensional parameter space and constantly replaces the vertices of the simplex using simple geometric rules based on the error values of the previous test points. As Nelder-Mead is essentially a gradient-based method, it may converge to local optima instead of the global optimum [53,72].

The *limited-memory Broyden-Fletcher-Goldfarb-Shanno algorithm with bound constraints (L-BFGS-B)* is considered to be a modern and efficient algorithm that aims to find a local minimum of the objective function using a limited amount of computer memory [73]. It is an iterative (quasi-Newton) algorithm that uses an efficient estimate of the inverse Hessian matrix to define the next search direction, executes a line search along the direction to find the next position, and checks for convergence.

**Single-objective global optimization algorithms.** The *Custom Evolutionary Optimization (CEO)* algorithm is a relatively simple member of the large class of evolutionary optimization algorithms. Evolutionary algorithms are metaheuristics for global optimization inspired by biological evolution. Each candidate solution, represented by a particular combination of the unknown parameters, is considered to be an individual within a population, and the value of the cost function for that parameter combination is treated as the "fitness" of that individual (with lower costs normally associated with higher fitness). The initial population typically consists of random samples from the search space. The population is then updated through the application of various operators. New individuals are generated via the application of genetic operators such as mutation, which introduces random variations into the parameters of an individual, and crossover, which randomly combines the parameters of two individuals. The size of the population is maintained by selecting individuals with higher fitness. These steps are repeated iteratively for a certain number of generations. Many different variants of evolutionary algorithms exist that differ in the details of the operators, and may also apply additional heuristics. The CEO algorithm is based on the EvolutionaryComputation class of the Inspyred package, and uses Gaussian mutation and blend crossover variators.

The *Classic Evolution Strategy (CES)* algorithm belongs to a subclass of evolutionary optimization algorithms called evolution strategies. In these algorithms, there are distinct mutation rates associated with each parameter, and these mutation rates are changed adaptively during the optimization [74]. In every iteration, all members of the parent population are used to create offspring, and the next generation is based on the best individuals from the combined (parents + offspring) population.

The *Covariance Matrix Adaptation Evolution Strategy (CMAES)* algorithm is an evolutionary algorithm which samples candidate solutions from multivariate normal distributions with adapting mean and covariance matrix [75]. CMAES samples a multivariate normal distribution to create a population, then implements selection and recombination as the evolutionary algorithms. Subsequently, it updates the covariance matrix from the current population and estimates the next step size for the evolution path. The algorithm itself is computationally expensive and less effective in case of hundreds of parameters [76]. We tested the original single-objective version of CMAES in this study, but a multi-objective version has also been developed and is also available in BluePyOpt.

The *Exponential Natural Evolution Strategy (XNES)* algorithm is an evolution strategy that uses the natural gradient to update the search distribution. The update rules in natural gradient updates of XNES are similar to CMAES. XNES calculates the expected fitness from natural gradient to change the multivariate Gaussian mutation distribution parameters [77].

The *Differential Evolution (DE)* algorithm is an evolutionary algorithm that generates new candidate solutions from existing individuals with a recombination operator based on a few

simple mathematical rules that also include some stochastic elements. The population is updated when candidates with higher fitness are found [52, 67].

The *Self-Adaptive Differential Evolution (SADE)* algorithm is a version of the Differential Evolution algorithm which adjusts the mutation rate and the crossover rate adaptively [78].

The *Particle Swarm Optimization (PSO)* algorithm represents candidate solutions as particles moving around in the search space. Each particle has a velocity and moves by adding this to its current position in every iteration. Initially the velocity is random, and it is modified after each iteration, influenced by the currently known best positions for the individual particles and that of the entire group. In this basic implementation, velocity and position updates are carried out sequentially for each particle [79].

The *Particle Swarm Optimization Generational (PSOG)* algorithm is similar to the PSO algorithm above but, in every iteration, it first updates the velocity for all particles, then updates the positions. This allows efficient parallel execution of the algorithm.

The *Extended Ant Colony Optimization (GACO)* algorithm is a bio-inspired algorithm based on the analogy of ants finding paths from the colony to food sources. In this algorithm, artificial agents move through the parameter space, and lay down "pheromones" depending on the quality of the solutions they find. These pheromones attract the other agents, making it more likely that they move to locations with high amounts of pheromone. This extended version of the algorithm calculates the locations of future generations of ants by sampling from a multi-kernel Gaussian distribution that depends on the quality of previously found solutions [80].

The *Basin-Hopping (BH)* algorithm is a generalization of the Simulated Annealing algorithm that was used in several earlier studies of neural parameter optimization [15, 21]. Basin-hopping is a two-level algorithm: its outer loop performs stochastic jumps in the search space, while the inner loop performs local optimization. The resulting new local minimum is always accepted if it is better than the previous one, but it may also be accepted if it is worse with a probability that depends on the increase in the cost function [81].

**Multi-objective global optimization algorithms.** The *Non-dominated Sorting Genetic Algorithm II (NSGA2)* is an evolutionary multi-objective algorithm. Multi-objective optimization algorithms aim to optimize several cost functions simultaneously, trying to find non-dominated (or Pareto-optimal) solutions where none of the cost functions can be improved without degrading the performance on some other cost functions. The algorithms also aim to create a diverse set of solutions that collectively provide good coverage of the Pareto front. In NSGA2, a child population is created from the parent population using the usual genetic operators, mutation and crossover. Individuals in the next generation are then selected from the joint population based on Pareto dominance and the so-called crowding distance that penalizes closely related individuals and helps maintain diversity within the population [68].

The *Pareto Archived Evolution Strategy (PAES)* algorithm is a simple multi-objective algorithm that uses local search (mutation) from the current individual(s) and maintains a reference archive of previously found non-dominated solutions to approximate the dominance ranking of candidate solutions [82].

The *Indicator Based Evolutionary Algorithm (IBEA)* is a multi-objective evolutionary algorithm that computes the fitness value based on predefined binary indicators. It performs environmental selection by removing the worst individuals, chooses parents by comparing the fitness values of randomly selected pairs of individuals, and applies mutation and crossover to create offspring, repeating the process iteratively until reaching the maximum number of generations [83].

The *Multi-objective Hypervolume-based Ant Colony Optimizer (MACO)* is a multi-objective optimization algorithm that extends the GACO algorithm described above, combining hypervolume computation and non-dominated fronts for ranking individuals [84].

The *Non-dominated Sorting PSO (NSPSO)* algorithm extends PSO by making a better use of personal bests and offspring for non-dominated comparison [85].

## Use cases

To compare the efficiency of various parameter search methods in solving neuronal parameter optimization tasks, we designed and implemented a suite of six different problems that may be considered typical use cases in this domain. All of these use cases can be handled by Neuroptimus, which allowed us to run all benchmarks using every selected algorithm within the same framework, and made the subsequent evaluation of their performance quite straightforward (see below). Five of the use cases were similar (or identical) to those presented in [15], although some of them were modified to increase the robustness of the simulations (avoiding errors due to invalid parameter combinations, in the case of the AdExpIF example) or to move the target behavior of the model away from a critical boundary (the transition to repetitive firing, in the case of the Hodgkin-Huxley model). We also describe a seventh use case that is outside the domain of tuning single-cell biophysical models based on electrophysiological target data. In this use case, the aim is to optimize the parameters of a model of the main biochemical cascades involved in long-term synaptic plasticity in the dendritic spines of pyramidal neurons. We provide a description of each use case below; all the files required to run these examples, along with detailed guides to setting up the optimizations in the Neuroptimus GUI, can be found in the corresponding subfolders of the neuroptimus/new_test_files directory of the Neuroptimus Github repository (https://github.com/KaliLab/neuroptimus).

**Use Case 1: Hodgkin-Huxley model.**   This use case is based on a single-compartment modelthat contains conductances from the original Hodgkin-Huxley model ($Na^+$, $K^+$, leak) [33], and is implemented in NEURON. To generate the target voltage trace, a suprathreshold step current was injected into the soma of the neuron model (amplitude = 300 pA, delay = 200 ms, duration = 500 ms, and the voltage trace duration is 1000 ms). The test case involves recovering the correct conductance densities (3 parameters) that were used to generate the target trace, while keeping the properties of the currents and the other parameters of the model constant (at their original values). A combination of four features (spike count, spike amplitude, spike width, mean squared error of voltage excluding spikes) was used to compare each simulated trace to the original (target) trace.

**Use Case 2: Voltage Clamp.**   In the Voltage Clamp benchmark problem, the same single-compartment model with the same conductances is used as in the Hodgkin-Huxley problem. In addition, this model contains a conductance-based synapse. The goal here is to recover the synaptic parameters (weight, rise and decay time constants, delay– 4 parameters) from simulated voltage clamp recordings during synaptic stimulation (four presynaptic spikes at 10 Hz), using the mean squared error cost function to compare the current traces.

**Use Case 3: Passive, anatomically detailed neuron.**   This benchmark uses a morphologically detailed passive model of a hippocampal CA1 pyramidal cell implemented in NEURON. During the experiment, a short (3 ms, 500 pA) and a long (600 ms, 10 pA) current pulse (separated by 300 ms) were injected into the soma, and the membrane potential was also recorded there. The neuron was filled with a dye during the recording and was reconstructed using Neurolucida. This reconstruction defines the morphology of the model, and the task involves fitting 3 passive parameters (specific capacitance, leak conductance density, specific axial resistance, all of which are assumed to be uniform within the cell) to reproduce the experimental data recorded using the same complex current clamp stimulus. Traces are compared via the mean squared error cost function. All the experimental data for this use case, including the

morphological reconstruction and the electrophysiological recordings, were provided by Miklós Szoboszlay and Zoltán Nusser.

**Use Case 4: Simplified active model.**    This use case attempts to fit the behavior of a six-compartmental simplification of a biophysically accurate and morphologically detailed hippocampal CA1 pyramidal cell model [40] to the somatic voltage responses of the original model with full morphology. Both models contained the same set of voltage-gated conductances in their somatic and dendritic compartments: transient Na channels (separate somatic and dendritic subtypes), delayed rectifier, A-type, and M-type voltage-gated K channels, C-type and AHP-associated Ca-dependent K channels, L-type and N-type Ca channels, and the hyperpolarization-activated HCN channels. Changes in the concentration of free calcium ions were also modeled separately in each compartment. Dendrites of the full model were clustered based on their passive voltage responses, and each of these clusters defined a dendritic compartment in the simplified model. The densities of ion channels in the dendritic compartments of the simplified model were set to the average values in the corresponding clusters of the full model, while the densities of the nine somatic conductances (somatic Na, proximal HCN, Ca (L), Ca(N), K(DR), K(A), K(M), K(C), K(AHP)) were subject to parameter optimization. The original full model was implemented in GENESIS, while the simplified model was implemented in the NEURON simulator. All model files, including the MOD files that implement voltage-gated channels and calcium concentration dynamics, are available in the subdirectory new_test_files/testcase_4_ca1_pc_simplification of the Neuroptimus GitHub repository. The target data was the voltage response of the full model to the injection of a 200 pA step current stimulus into the soma (the stimulus started at 200 ms and lasted for 600 ms, with a total recording duration of 1000 ms). The fit was evaluated via a combination of features including mean squared error (excluding spikes) weighted by 0.2, spike count (weight 0.4), latency to first spike (weight 0.1), action potential amplitude (weight 0.1), action potential width (weight 0.1), and after-hyperpolarization depth (weight 0.1).

**Use Case 5: Extended integrate-and-fire model.**    In this benchmark problem, the parameters of a phenomenological (adaptive exponential integrate-and-fire) spiking model [37, 38], implemented in the NEST simulator [39] were fitted to capture the somatic responses of a real neuron (hippocampal CA3 pyramidal cell) to four different inputs. Voltage traces were recorded experimentally in response to current steps of 900 ms duration, and 0.30, 0.35, 0.40, and 0.45 nA amplitudes (the step was delayed by 100 ms, and the recordings lasted for 1100 ms). The sampling frequency was 5 kHz. The unknown parameters to be optimized were the capacitance, the leak conductance, the reversal potential of the leak current, the threshold voltage, the reset voltage, the refractory period, the steepness of the exponential part of the current-voltage relation, the subthreshold adaptation conductance, the spike adaptation current, and the adaptation time constant (10 parameters). During the optimization the mean squared error (excluding spikes), the spike count (during stimulus), and the latency to first spike error functions were used with equal weights.

**Use Case 6: Morphologically and biophysically detailed CA1 pyramidal cell model.**
This is our most complex benchmark problem both regarding the number of parameters to be optimized and the complexity of the model. The test case is based on an anatomically and biophysically detailed rat hippocampal CA1 pyramidal cell model built for the NEURON simulator in our research group. The morphology of the model was from [44]. The model contained several different voltage-gated ion channels in its somatic, dendritic, and axonal compartments: a transient Na conductance, delayed rectifier, A-type, M-type, and D-type voltage-gated K conductances, and the hyperpolarization-activated current Ih. Many attributes of the model were well-constrained by experimental observations available in the literature, including the distributions and kinetic properties of the ion channels. The target data (provided by

Judit Makara) consisted of the means (and associated standard deviations) of 20 different types of features extracted by eFEL from the voltage responses of five rat CA1 pyramidal neurons to somatic current step injections of six different amplitudes (-0.25, 0.05, 0.1, 0.15, 0.2 0.25 nA), with each stimulus repeated three times for every cell. This resulted in a total of 66 feature values to be matched by the model. The eFEL features and the associated current step amplitudes are listed in S3 Table, and some examples of the original patch-clamp recordings are shown in S7 Fig.

We optimized 12 abstract parameters of the model that were mapped onto the actual parameters of the NEURON implementation by an appropriate user function. Ten parameters determined the densities of the voltage-gated and leak conductances in the different compartments (soma, dendrites, axon) of the neuron. One of these represented the somatic density of the Na conductance, which also indirectly determined the density of axonal and dendritic conductances via fixed multipliers; the dendritic multiplier also depended on the distance from the soma. Three parameters directly determined the density of K(DR) in the soma, dendrites and axon, respectively. One parameter set the density of K(M) in the soma and the dendrites, and there was another parameter for the axonal K(M). We used a scaling parameter for K(A) that multiplied the non-uniform baseline density everywhere in the neuron, and applied to both proximal and distal versions of the conductance. The conductance of HCN channels was set by another scaling parameter using a very similar strategy. One parameter set the (uniform) density of the leak conductance, and another one determined the density of K(D), which was present only in the soma. One parameter represented the reversal potential of the leak current; and the final parameter determined the difference between the half-activation and half-inactivation potential values of the Na conductance. All model files, including the MOD files that implement voltage-gated channels and the text file containing the user-defined function for setting the parameters, are available in the subdirectory new_test_files/testcase_6_Detailed_-CA1_pyramidal_cell_model of the Neuroptimus GitHub repository.

**Use Case 7: Intracellular biochemical pathway model.** In this use case, we fit the input fluxes and the initial concentrations of the main molecular species of an intracellular biochemical network model describing three cascades (CaMKII, PKA, and PKC) that shape postsynaptic plasticity [45]. The model contains 204 different molecular species including the complexes and distinct forms of particular biomolecules. There are 261 reactions and 412 reaction rates that describe the actions of the molecules and the interactions between them. The model has 4 inputs; the main input is the NMDA receptor-mediated calcium ion flux, which is modeled as a direct influx into the spine head. The other 3 input fluxes: the glutamate, acetylcholine, and β-adrenergic ligand fluxes activate their specific metabotropic receptors, leading to the production of different second messengers (cAMP, DAG, IP$_3$), which, along with the free calcium ions, activate different kinases and phosphatases. The CaMKII, PKA, and PKC kinases phosphorylate, and the PP1, PP2B, and PP2A phosphatases dephosphorylate the GluR1 and GluR2 AMPA receptor subunits. The phosphorylation state of these AMPA receptor subunits determines the localization and the single-channel conductance of the receptor tetramers. The output of the model is the total synaptic conductance of the membrane-bound AMPA receptor population, which is determined by their number, composition, and phosphorylation state, estimated by a statistical model [45].

The use case involves the fitting of 12 parameters including three input fluxes: calcium ('Caflux'), glutamate ('Gluflux'), and β-adrenergic ligand ('Lflux'); the ratio of the GluR1 and GluR2 subunits ('GluR1_ratio'); and 8 parameters that weight the main molecular species of the model. These weight parameters (parameter names starting with 'IC_' in Fig 8E) set the initial concentrations of the related molecular species with a multiplying factor between 0 and 4, and correspond to the factor parameters described in [45]. 'IC_PLC', 'IC_AC', 'IC_CaMCK',

'IC_PP', 'IC_PDE' correspond to $f_{PKC}$, $f_{PKA}$, $f_{CaMKII}$, $f_{PP}$, and $f_{PDE}$, respectively. Besides, 'IC_NCX', 'IC_PKA', and 'IC_PKC' set the initial concentrations of NCX, PKA, and PKC proteins, respectively. A more detailed description of the parameters can be found in the 'Parameter alterations and model fitting' section of [45]. All model files are available in the subdirectory new_test_files/testcase_7_Biochemical_pathway_model of the Neuroptimus GitHub repository.

We selected an experimental data set from [45] that includes recordings from entorhinal cortical neurons under two experimental conditions (originally published in [47]), and used it as the target data to fit the model by optimizing its parameters. The first part of the data set is the time course of peak EPSP responses following LTP induction using a typical high-frequency stimulation (HFS) protocol (100 Hz, 1 s). The second part of the data set (also peak EPSP time course) corresponds to an experiment in which PKA was inhibited, and its contribution to HFS-induced LTP was investigated. The peak EPSP time courses were sampled in 2-minute intervals from the time point of the stimulation and were compared with the relative synaptic conductance produced by the model at the corresponding time points. Then the mean of the squares of the errors (the differences between the predicted relative conductance values and the corresponding relative peak EPSP values at each sampled time point) was calculated as the error function. We set the population size to 100, and the number of generations to 100 as well, and ran the optimizer using three different algorithms with default parameter settings: CES (Inspyred), CMAES (Cmaes), and PSO (Inspyred). We chose these algorithms because they achieved the best results in the previous use cases and are suitable for parallelization. We ran 10 parallel optimizations with each algorithm on the Neuroscience Gateway (NSG).

## Evaluation Methods

We tested the different optimization algorithms on each of the first six model optimization tasks described above (Use Case 7 was run only with a subset of the algorithms due to its high demand for computing power). To ensure a fair comparison of model performance, we allowed 10,000 model evaluations for every algorithm on each task. For all population-based methods (including evolutionary algorithms and swarm intelligence-based approaches) we set the population size to 100, and the number of generations to 100 as well. We used this split because it represented a reasonable balance between having large enough populations and a sufficient number of generations; it also allowed an efficient use of computational resources because the supercomputers that we used via NSG had 128 cores per node, and thus a whole population of 100 could be evaluated in parallel on these machines. We similarly set 100 global and 100 local minimization steps for two-stage algorithms. Otherwise, we ran every algorithm with its default settings in Neuroptimus. These default options are typically the package default settings, with one significant exception: we observed that the default settings of the EvolutionaryComputation class of the Inspyred package that underlies our CEO algorithm led to essentially no optimization, so we adjusted the default number of elites from 0 to half of the population size, changed the mutation rate from 0.1 to 0.25 and the standard deviation of Gaussian mutation from 1 to 0.5.

Optimization runs were parallelized for all algorithms where this is supported by Neuroptimus and the underlying packages (see Table 2). For the most resource-intensive use cases (the detailed CA1 pyramidal neuron model and the biochemical pathway model) these parallelized runs were performed on supercomputers via the Neuroscience Gateway [86]; simpler use cases and algorithms that do not support parallelization were run on a Dell PowerEdge R730 compute server or personal computers. To allow meaningful statistical comparisons between the algorithms, we performed 10 independent runs (using distinct random seeds) of each algorithm in every use case.

We visualized and compared the performance of the algorithms in each use case using several different methods. All the comparisons were based on the change in the total error during the optimization. First, we visualized the convergence of the algorithms by plotting the cumulative minimum of the error function after every generation (i.e., after every 100 model evaluations). We plotted the median value across the 10 runs to see which algorithms typically find the best solutions after a given number of model evaluations. The lowest and the highest errors achieved by the 10 runs were also calculated in every iteration to observe how well the algorithm performs in the best case, and whether it gets stuck in some cases.

We defined two basic scores to characterize and compare the performance of the algorithms in a concise manner. The first of these scores was defined as the lowest error achieved during the entire optimization run (these are usually, but not always, associated with members of the final population). We visualized the distribution of this measure across the 10 independent runs using box plots that show the median, interquartile range, minimum, and maximum values, and also indicate apparent outliers.

In the case of more complex, detailed models, each model evaluation (simulation) can be time-consuming, and thus we are also interested in which algorithms can find a reasonably good solution in a relatively short time. To characterize the convergence speed of an algorithm, we used the sum of the logarithms of the error scores achieved by the best individuals in each generation. This is essentially the area under the logarithmic convergence curve—the smaller this sum is, the faster the algorithm found a relatively good solution.

## Supporting information

**S1 Fig. Evolution of the individual error components during the optimization of the Hodgkin-Huxley model using the CMAES algorithm.** The curves show the median of 10 independent runs. Each generation corresponds to 100 model evaluations.
(TIF)

**S2 Fig. Voltage trace of the target data and the traces corresponding to the best parameters found by the Covariance Matrix Adaptation Evolution Strategy (CMAES) and Random Search algorithms in Use Case 1 (Hodgkin-Huxley model).**
(TIF)

**S3 Fig. Current trace of the target data and the traces corresponding to the best parameters found by the Covariance Matrix Adaptation Evolution Strategy (CMAES) and Random Search algorithms in Use Case 2 (voltage clamp).**
(TIF)

**S4 Fig. Voltage trace of the target data and the traces corresponding to the best parameters found by the Covariance Matrix Adaptation Evolution Strategy (CMAES) and Random Search algorithms in Use Case 3 (passive, anatomically detailed neuron).**
(TIF)

**S5 Fig. Voltage trace of the target data and the traces corresponding to the best parameters found by the Covariance Matrix Adaptation Evolution Strategy (CMAES) and Random Search algorithms in Use Case 4 (simplified active model).**
(TIF)

**S6 Fig. Voltage trace of the target data and the traces corresponding to the best parameters found by the Covariance Matrix Adaptation Evolution Strategy (CMAES) and Random Search algorithms in Use Case 5 (extended integrate-and-fire model).**
(TIF)

**S7 Fig. Examples of somatic current-clamp recordings from 5 different CA1 pyramidal cells (columns, with the individual cell IDs) to 6 current step amplitudes (rows).** For each cell the current step protocol was repeated 3 times, and the figure shows the results of the first recordings from the cells. Features extracted from these experimental data (5 cells and 3 repeated recordings, which resulted in 15 recordings) were used as the target data in Use Case 6 (morphologically and biophysically detailed CA1 pyramidal cell model). (TIF)

**S8 Fig. This figure shows a comparison between the feature values extracted from somatic current-clamp recordings from CA1 pyramidal neurons (see S7 Fig for examples) and the features resulting from simulations of the best solutions found by the CMAES algorithm and the random search algorithm.** The experimental mean and standard deviation are represented by the black cross and error bar, while the red and blue dots correspond to the feature values in simulations of the best parameter combinations resulting from 10 independent runs of the CMAES and random search algorithms, respectively. Values along the vertical axis denote the amplitude of the current injection used. Note that the feature values from the CMAES solutions are typically closer to the experimental mean values, and have smaller variability, than those resulting from random search. (TIF)

**S9 Fig. Overall rankings of single-objective optimization algorithms.** Statistics of the ranks achieved by single-objective optimization algorithms on the six different benchmarks (Figs 1–6) according to the final error (A) and convergence speed (B). Brown dots represent the ranks achieved by the algorithms in each use case; boxes indicate the full range and the orange line represents the median of these ranks. The color of the name of the algorithm indicates the implementing package, with the color code included in the legend. Algorithms are sorted according to the median of their ranks. (TIF)

**S1 Table. Error component scores corresponding to the best parameters found by the Covariance Matrix Adaptation Evolution Strategy (CMAES) and Random Search algorithms in Use Case 4 (simplified active model).** The features used for this problem are mean squared error (excluding spikes), spike count, latency to first spike, action potential amplitude, action potential width, and after-hyperpolarization depth. (XLSX)

**S2 Table. Names and values of hyperparameters and other settings used for each algorithm in this article.** (XLSX)

**S3 Table. List of eFEL features (with brief explanations) and associated current amplitudes used as the target data in Use Case 6 (morphologically and biophysically detailed CA1 pyramidal cell model).** (XLSX)

## Author Contributions

**Conceptualization:** Máté Mohácsi, Szabolcs Káli.

**Funding acquisition:** Szabolcs Káli.

**Investigation:** Máté Mohácsi, Márk Patrik Török, Sára Sáray, Luca Tar, Gábor Farkas.

**Methodology:** Máté Mohácsi, Sára Sáray, Szabolcs Káli.

**Software:** Máté Mohácsi, Márk Patrik Török, Sára Sáray, Luca Tar, Gábor Farkas.

**Supervision:** Szabolcs Káli.

**Validation:** Luca Tar, Gábor Farkas.

**Visualization:** Máté Mohácsi.

**Writing – original draft:** Máté Mohácsi, Szabolcs Káli.

**Writing – review & editing:** Máté Mohácsi, Márk Patrik Török, Sára Sáray, Luca Tar, Gábor Farkas, Szabolcs Káli.

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
