## [Decision Letter · Decision Letter 0]

27 Jul 2024

Dear Dr. Káli,

Thank you very much for submitting your manuscript "Evaluation and comparison of methods for neuronal parameter optimization using the Neuroptimus software framework" for consideration at PLOS Computational Biology.

As with all papers reviewed by the journal, your manuscript was reviewed by members of the editorial board and by several independent reviewers. In light of the reviews (below this email), we would like to invite the resubmission of a significantly-revised version that takes into account the reviewers' comments.

After thorough evaluation by our reviewers, we have concluded that major revisions are required before your manuscript can be considered for publication. Many strengths were identified in the study, including integration of several standard open-source optimization packages, providing both an API and a GUI, offering valuable guidelines and recommendations for users, and the interactive website for sharing optimizations. However, several key points need to be addressed, including:

1) The introduction needs more context on parameter optimization challenges, a clear research gap, and a detailed rationale for Neuroptimus.

2) The methods section lacks sufficient detail on the specific algorithms compared, the benchmark scenarios used, and the evaluation metrics.

3) The results and discussion sections require additional comparisons with existing literature, a discussion on the strengths and limitations of each algorithm, and practical implications of the findings.

We cannot make any decision about publication until we have seen the revised manuscript and your response to the reviewers' comments. Your revised manuscript is also likely to be sent to reviewers for further evaluation.

Sincerely,

Salvador Dura-Bernal

Guest Editor

PLOS Computational Biology

Andrea E. Martin

Section Editor

PLOS Computational Biology

After thorough evaluation by our reviewers, we have concluded that major revisions are required before your manuscript can be considered for publication. Many strengths were identified in the study, including integration of several standard open-source optimization packages, providing both an API and a GUI, offering valuable guidelines and recommendations for users, and the interactive website for sharing optimizations. However, several key points need to be addressed, including:

1) The introduction needs more context on parameter optimization challenges, a clear research gap, and a detailed rationale for Neuroptimus.

2) The methods section lacks sufficient detail on the specific algorithms compared, the benchmark scenarios used, and the evaluation metrics.

3) The results and discussion sections require additional comparisons with existing literature, a discussion on the strengths and limitations of each algorithm, and practical implications of the findings.

Reviewer's Responses to Questions

**Comments to the Authors:**

Reviewer #1: This is a paper describing a new tool Neuroptimus, to optimize parameters of neuroscientific models. It is based on a previous tool developed previously by some of the authors. The paper also describes results obtained with the tool. Namely the comparison of different optimization algorithms to some benchmark scientific problems.

I will start with a general remark I have concerning this types of papers. The authors also point it out partly in their discussion. But I'm always reluctant wrt the type of papers where creators of a software/algorithm make one-sided comparisons between code retrieved from other sources. Maybe the situation is slightly different here, since all the code they test is available as an option in their software. The problem with search algorithms is that the very specific implementation, and especially the hyperparameters can be tremendously important for comparisons. And it will always depend on the effort of the people trying the algorithms to pick the best parameters/implementation.

So in general, and this will come back later, this should be stated as clearly as possible. I would even say in the abstract.

Then some more specific comments:

Line 138: is it possible to e.g. provide constraints to the parameters. The obvious constraint is bounds, which I assume is possible. But what about other linear/nonlinear constraints?

Wrt to this I also have general remark. Can you maybe describe how flexible your software setup is, especially wrt to the GUI. GUI's obviously make things easier for many users, but they also tend to put software in a very strict input schema. Did you e.g. take any special measures to make sure users can easily extend the GUI conceptually. (With e.g. special constraints as an example).

Line 201: Here it might be useful to cite some relevant papers that discuss surrogate data. One I can think of is Druckmann 2008

Line 241: Related to my hyperparameter comment above, did you in any way make sure that the hyperparameters of the algorithms you used were tuned for this 10k number of runs? As an example, which i'm sure you didn't do, but if e.g. one would pick a evolutionary algorithm with a 10k population size, this would obviously be bad, which is an extreme case. But many algorithms have hyperparameters that need to be tuned based on the input problem.

Fig 1 (and further figures). I think it would be good to (maybe in supp material?) in some way show the evolution of the subfeatures the score consists of. The total score give one view of the problem, but for a user it can be important not to just a solution with a low total score, but also a score that is low for all features at the same time as opposed to a solution with one feature being extremely low, but other still being high.

And related to this a more general remark. I'm not saying to redo your entire analysis, but I think it's necessary to at least mention that the measure you use is an extremely simple one, and might not convey all the metrics that a user might be interested in. Some I can think of are: diversity of the final solution population (e.g. a user might want to build network for which a set of solutions might be required, instead of just the best one), weighing specific features and how well the algorithm follows these weights.

It would actually be good to give a table with the capabilities of all the algorithms wrt to this. E.g. 'is' there even a population in the algorithm, is it MO, etc etc.

And also in how far it allows parallelization. Some algorithms inherently are good to parallelize, for others it might be virtually impossible. And this can be extremely relevant wrt to speed, the 'number of evaluations necessary' is not the only metric relevant for the speed.

Fig 2: it seems there are some missing methods wrt to the other figures, i'm not sure i saw a reason for that somewhere.

Line 372: Could you be a bit more specific about 'orders of magnitude' here, or what this means wrt to the actual resulting traces. Sometimes getting very close to the target trace can be perceived as overfitting the data so irrelevant. So it might be good to show what these differences mean in practice.

Fig 6A: Can you please also show the spread of the underlying data here, if necessary in supp material. It would be good to get an idea on how well these fits match with the actual raw traces.

Line 482: So i assume you used SO CMAES? Afaik CMA-ES also has a multi-objective variant, might be at least worth mentioning.

Line 645: One thing I'm a missing in the text is a mention that the choice of exact operators in the algorithm can be very important too, apart from hyperparameters (unless I missed it). E.g. evolutionary algorithm have a wide choice of mutation, crossover operators, elitism, etc. which can heavily influence the outcome. This is again a reason why comparing algorithms based on just names is extremely dangerous, since each algorithm name corresponds with an entire ecosystem of variants.

Line 864: I'm curious if the GUI enables things like restarting from existing checkpoints etc.

Line 993: I just wonder if it's really necessary to include all these short descriptions of these algorithms instead of pointing to paper (and maybe have a 1 sentence description in a table or so).

Line 1110: We don't really see these traces anywhere I think. It would be good that we at least have some links to methods that were used to generate this experimental traces.

Reviewer #2: The authors present a new Python-based software package called Neuroptimus, that contains an API

and GUI for performing neural model parameter optimization through several standard open-source

optimization packages. The authors test the different optimization packages on several optimization problems

of increasing complexity with the aim to provide guidelines for users with similar optimization

problems. They derive several recommendations, in particular that CMAES is one of the better

algorithms. The output of their optimizations are provided for the community, as well as a website

for uploading additional user-defined optimizations. The interactivity through the website is

an added benefit that could grow the use of the package. Most of the paper focuses around single

neuron optimization. Although this has been extensively studied in the past, collating multiple

libraries under Neuroptimus, may help the community leverage the experience the authors have gained

through their extensive tests.

One drawback of the focus on single neuron optimization is that many of the issues have been more

or less solved. This would therefore limit the scope/relevance of use of their new package. For that

reason, I would like to see some additional studies (though certainly not with the same depth/breadth

as for single neuron optimization) on neuronal network optimization and sub-cellular optimization (e.g. RxD).

The authors mention that they have already performed these specific studies in the Discussion.

Including the details and results of those studies would increase the appeal of their work and

allow a wider audience to benefit. In addition, it would showcase how to use their software

for computational optimization tasks that are less "solved", or generally solvable with more effort

than single neuron optimization which has a broad literature spanning decades.

Towards this end of network optimization, I would suggest highlighting several recent papers that used

evolutionary strategies to fit model parameters to reproduce experimental data:

Data-driven multiscale model of macaque auditory thalamocortical circuits reproduces in vivo dynamics

or to teach models to perform behaviors through Evolutionary Strategy:

Training spiking neuronal networks to perform motor control using reinforcement and evolutionary learning

These approaches could form the basis of comparison for a network based approach to optimization

through Neuroptimus.

Detailed comments

abstract:

"Recently, manual model tuning has been replaced by automated parameter search using a variety of different tools and methods."

"Recently" - automated parameter optimization has been used for decades

Introduction:

Left out an important, recent paper on optimizing single neuron models to experimental data using

evolutionary multiobjective optimization:

Journal of Neurophysiology: 117(1):148-162, 2017

The study is well done and interesting but the authors claim in several places that the most widely

performed optimization in neuroscience is on the single neuron level. This may have been true in the

past, but lately, a lot of recent state-of-the-art work has been done on optimizing detailed neuronal

circuit models to replicate the patterns seen in electrophysiological data, by several different groups:

Data-driven multiscale model of macaque auditory thalamocortical circuits reproduces in vivo dynamics

Multiscale model of primary motor cortex circuits predicts in vivo cell-type-specific, behavioral state-dependent dynamics

Reconstructing visual experience from a large-scale biologically realistic model of mouse primary visual cortex

Uncovering circuit mechanisms of current sinks and sources with biophysical simulations of primary visual cortex

HNN-core: A Python software for cellular and circuit-level interpretation of human MEG/EEG

Although it may be beyond the scope of the current work to include network/circuit-based optimization,

not even discussing this latest trend on a paper on neuronal optimization, seems to be missing

an important part of current reseearch. Can the authors include more discussion of network optimization?

Alternatively, it would be useful if they can extend their software to include network optimization

on a simple data-set. At least this would broaden the audience for their software and keep it relevant.

Previous work mentioned above (Neymotin 2017) uses a step-wise approach to optimizing conductance-based neuronal models.

The first stage involves fitting the passive parameters (leak conductance, capacitance, etc.) to subthreshold

current injections. Afterwards active channel conductances contributing to spike generation are fitted

to reproduce accurate firing patterns (rate vs injected current at different steps, action potential

shape, etc.). In that study, the authors found better performance using this strategy. Can the authors

provide a similar step-wise comparison instead of optimizing all functions at once? Or if this is

not possible in general, can they comment on effective strategies for dealing with these types

of complexities?

Discussion line 554: the authors mention only NEURON supports a GUI. That is not correct, as far as I know

NetPyNE and Human Neocortical Neurosolver both have GUIS and both support optimization of neuronal

network models.

If CMAES and PSO are the best algorithms, why include all the others? Were any of the others best

in certain circumstances? While claiming that many algorithms are a benefit to the users, naive

users will not know which algorithm to select. Perhaps having clear default algorithms and parameters

in the GUI and code-base would make it easier for the novice user to find the algorithmic

equivalent of the "needle in the haystack". Can the authors comment on this tradeoff between

large number of algorithms vs best algorithms and how users might be encouraged to use

the best algorithms by default?

Possible Extensions in Discussion is an interesting section that mentions use of Neuroptimus

outside the scope of single cell neuronal modeling. I would like to see some of the results

of the the NEURON RxD optimization simulation and the network model in particular, since it broadens

the scope of the software for more interesting/timely research questions, even if not including

a comparison of all optimization algorithms. Without these examples, the tool may not be

as attractive to the research community. Note that the optimizations and network models

describe above could have benefitted from a tool such as the one described in the manuscript.

By only focusing on single cell models, the authors detract from potential users of the

software.

Methods

The authors mention models should generally be implemented in HOC. HOC is a bit outdated, and most

new NEURON models are specified in Python. Can the authors clarify if Python implementations of

neuronal models are supported?

Reviewer #3: 1. The introduction sets the stage for the study but could be strengthened by:

Providing more context on the challenges of parameter optimization in computational neuroscience.

Clearly stating the research gap or problem that the study aims to address.

Including a more detailed rationale for the development of the Neuroptimus software framework and its potential impact on the field.

2. Methods: The methods section lacks sufficient detail on the specific algorithms compared, the benchmark scenarios used, and the parameters for optimization. To enhance the methods section, the authors could consider providing a clear description of each algorithm tested, including any modifications or adaptations made. The authors should also detail the benchmark scenarios used, explaining their relevance to neuronal parameter optimization and clearly outline the evaluation metrics used to compare the performance of the algorithms.

3. Results: In lines 1066-1084, the authors describe some dendritic features and channel distributions. However, the authors should describe how Neuroptimus fares with extracting dendritic parameters such as the attenuation of back-propagating action potentials and PSP amplitudes as a function of dendritic section distance from the soma.

4. Discussion: Suggestions for improvement in the discussion section include:

Discussing the strengths and limitations of each algorithm tested to the benchmark scenarios.

Comparing the findings with existing literature on neuronal parameter optimization methods.

Highlight the practical implications of the results for researchers in computational neuroscience.

**Have the authors made all data and (if applicable) computational code underlying the findings in their manuscript fully available?**

Reviewer #1: None

Reviewer #2: Yes

Reviewer #3: Yes

PLOS authors have the option to publish the peer review history of their article (what does this mean?). If published, this will include your full peer review and any attached files.

Reviewer #1: **Yes: **Werner Van Geit

Reviewer #2: No

Reviewer #3: No
---

## [Decision Letter · Decision Letter 1]

14 Nov 2024

Dear Dr. Káli,

We are pleased to inform you that your manuscript 'Evaluation and comparison of methods for neuronal parameter optimization using the Neuroptimus software framework' has been provisionally accepted for publication in PLOS Computational Biology.

Best regards,

Salvador Dura-Bernal

Guest Editor

PLOS Computational Biology

Andrea E. Martin

Section Editor

PLOS Computational Biology

Feilim Mac Gabhann

Editor-in-Chief

PLOS Computational Biology

Jason Papin

Editor-in-Chief

PLOS Computational Biology

Reviewer's Responses to Questions

**Comments to the Authors:**

Reviewer #1: In my opinion the authors have sufficiently addresses the comments of the reviewers in the new manuscript.

Reviewer #2: Thank you for addressing my concerns.

Reviewer #3: The authors have addressed all my concerns in the revised manuscript.

**Have the authors made all data and (if applicable) computational code underlying the findings in their manuscript fully available?**

Reviewer #1: Yes

Reviewer #2: Yes

Reviewer #3: Yes

PLOS authors have the option to publish the peer review history of their article (what does this mean?). If published, this will include your full peer review and any attached files.

Reviewer #1: **Yes: **Werner Van Geit

Reviewer #2: No

Reviewer #3: No

---

## [Editor Report · Acceptance letter]

28 Nov 2024

PCOMPBIOL-D-24-00560R1 

Evaluation and comparison of methods for neuronal parameter optimization using the Neuroptimus software framework

Dear Dr Káli,

I am pleased to inform you that your manuscript has been formally accepted for publication in PLOS Computational Biology. Your manuscript is now with our production department and you will be notified of the publication date in due course.

With kind regards,

Anita Estes
